# A Validation Approach to Over-parameterized Matrix and Image Recovery

Lijun Ding [*]   Zhen Qin [†]   Liwei Jiang[‡]   Jinxin Zhou [†]   Zhihui Zhu [†]

This paper studies the problem of recovering a low-rank matrix from several noisy random linear measurements. We consider the setting where the rank of the ground-truth matrix is unknown a priori and use an objective function built from a rank-overspecified factored representation of the matrix variable, where the global optimal solutions overfit and do not correspond to the underlying ground truth. We then solve the associated nonconvex problem using gradient descent with small random initialization. We show that as long as the measurement operators satisfy the restricted isometry property (RIP) with its rank parameter scaling with the rank of the ground-truth matrix rather than scaling with the overspecified matrix rank, gradient descent iterations are on a particular trajectory towards the ground-truth matrix and achieve nearly information-theoretically optimal recovery when it is stopped appropriately. We then propose an efficient stopping strategy based on the common hold-out method and show that it detects a nearly optimal estimator provably. Moreover, experiments show that the proposed validation approach can also be efficiently used for image restoration with deep image prior, which over-parameterizes an image with a deep network.

## 1. Introduction

We consider the problem of recovering a low-rank positive semidefinite (PSD) ground-truth matrix $X_\natural \in \mathbb{R}^{n \times n}$, a symmetric matrix with all its eigenvalues nonnegative, of rank $r_\natural$ from $m$ many *noisy* linear measurements:

$$y = \mathcal{A}(X_\natural) + e, \tag{1}$$

where $\mathcal{A} : \mathbb{R}^{n \times n} \to \mathbb{R}^m$ is a known linear measurement operator, and $e \in \mathbb{R}^m$ is the additive independent noise with subgaussian entries with a variance proxy $\sigma^2 \geq 0$. We denote the dataset by $(y, \mathcal{A})$.

Low-rank matrix recovery problems of the form (1) appear in a wide variety of applications, including quantum state tomography, image processing, multi-task regression, metric embedding, and so on [1–5]. A computationally efficient approach that has recently received tremendous attention is to factorize the optimization variable into $X = UU^{\mathrm{T}}$ with $U \in \mathbb{R}^{n \times r}$ and optimize over the $n \times r$ matrix $U$ rather than the $n \times n$ matrix $X$ [5–16]. This strategy is usually referred to as the matrix factorization or the Burer-Monteiro approach in [17, 18]. We refer to the parameter $r$ as the *parametrized rank*. With this parametrization of $X$, we recover the low-rank matrix $X_\natural$ by solving

$$\underset{U \in \mathbb{R}^{n \times r}}{\text{minimize}} \ f(U) := \frac{1}{2m} \left\| \mathcal{A}\left(UU^\top\right) - y \right\|_2^2. \tag{2}$$

**Overparametrization: definition and its necessity** We refer to the parametrization $X = UU^\top, U \in \mathbb{R}^{n \times r}$ with $r > r_\natural = \mathrm{rank}(X_\natural)$ as *overparametrization* and the case of $r = r_\natural$ as exact parametrization. When $r = r_\natural$, it is well-known that simple gradient descent methods can find a matrix with a statistical error that is minimax optimal up to log factors [5, 8, 9]. However, in practice, the ground-truth

---

[*]Department of Mathematics, University of California San Diego, La Jolla, CA, USA (`l2ding@ucsd.edu`.)

[†]Department of Computer Science and Engineering, Ohio State University, Columbus, OH, USA (`qin.660@osu.edu`, `zhou.3820@osu.edu`, `zhu.3440@osu.edu`).

[‡]H. Milton Stewart School of Industrial and Systems Engineering, Georgia Institute of Technology, Atlanta, GA, USA (`ljiang306@gatech.edu`).

Second Conference on Parsimony and Learning (CPAL 2025).

rank $r_\natural$ is usually *unknown a priori*, and it can be challenging to identify the rank $r_\natural$ precisely. Fortunately, one can use overparametrization, since an upper bound of $r_\natural$ is often available from domain expertise [5, 19] or abundant computational resources. Apart from this practical concern, it is also theoretically interesting to study overparametrization of the Burer-Monteiro approach due to its connection to modern neural networks, which are almost always overparametrized [20–22]. These two concerns have induced a vibrant research direction recently [21, 23–29]. Below, we discuss two primary issues raised by overparametrization in the noisy regime ($\sigma > 0$) that the existing literature [23, 24, 29] may not adequately address.

**Overparemetrization: overfitting issue** Overparametrization introduces more parameters to estimate, hence the classical issue of machine learning: overfitting. Following [23], in Figure 1,[4] We implement a gradient descent (GD) method coupled with the so-called spectral initialization (which enables $U_0 U_0^\top \approx X_\natural$) for (2) proposed in [23] with output $\hat{U}$ and the recovered matrix $\hat{X} = \hat{U}\hat{U}^\top$, and recorded the recovery error (or the statistical error), $\|\hat{X} - X_\natural\|_F^2$ and the training error, $f(\hat{U})$. The recovery error (solid black line) increases as the estimated rank $r$ becomes larger than $r_\natural$, while the training error (dashed black line) decreases. This observation is due to the noise being overfitted and is consistent with the guarantees developed in [23, 24, 29], whose bound on the recovery error scales at least linearly in the estimated rank $r$. However, it is known that the convex approach [30] produces an estimator with a statistical error that scales linearly with the true rank $r_\natural$, and is also nearly minimax optimal [2, Theorem 2.5].

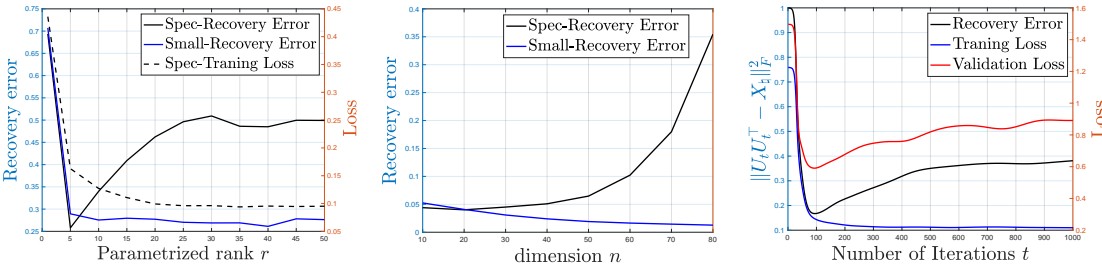

(a) Recovery error and Loss vs parametrized rank

(b) Recovery error vs dimension $n$

(c) Training vs Recovery vs Validation

Figure 1: (a) In Figure 1a, the black solid and dashed line represents the recovery error $\|\hat{X} - X_\natural\|_F$ and the training loss $f(\hat{U})$ respectively for the estimator $\hat{X} = \hat{U}\hat{U}^\top$ given in [23] when the parametrized rank $r$ varies from 0 to 50. The blue solid shows the recovery error given by our estimator. Here, $n = 50, m = 1000, \sigma = 0.3$, and $r_\natural = 5$. (b) In Figure 1b, the black line and the blue line show the recovery error of the estimator given in [23] and this paper, respectively. Here $m = nr_\natural$, $\sigma = 1/n, r_\natural = 5$, and $n$ varies from 10 to 80. (c), Gradient descent for the over-parameterized matrix recovery (2) with $r = n$: training loss (blue curve) is $f(U_t)$ in (2), and the validation loss (red curve) is the same loss but on the validation data.[4]

**Overparemetrization: stronger condition issue on** $\mathcal{A}$ Apart from overfitting, another issue for existing analysis of gradient descent methods coupled with spectral initialization for solving (2) as those in [23, 24, 29] is that the operator $\mathcal{A}$ has to satisfy $2r$-*RIP* (see Definition 2.1 for the formal definition). In general, the larger $r$ is, the stronger the condition is, and the more measurements are needed to ensure $\mathcal{A}$ satisfying $2r$-RIP [30]. However, it is known that for the convex approach, only $2r_\natural$-RIP is needed [30]. We note that the requirement on $\mathcal{A}$ is not merely a theoretical defect. In Figure 1b, we set the parametrized rank $r = n$, the noise parameter $\sigma = 1/n, m = 4nr_\natural$, and perform the method developed in [23].[4] Note that the theoretical recovery error in [23], $\tilde{\mathcal{O}}(\sigma^2 nr/m) = \tilde{\mathcal{O}}(1/nr_\natural)$ [5] with the above choice of $\sigma$, $m$, and $r$. However, we can see that the error (the black line) increases as the dimension $n$ increases, meaning that $\mathcal{A}$ fails the $2r$-RIP.

---

[4]Detailed descriptions of the experiments in Figure 1 are in Section B.
[5]The notation $\tilde{\mathcal{O}}(\cdot)$ hides logarithmic factors of $n, r, \sigma$ and the dependence of the condition number of $X_\natural$.

Table 1: Comparison with prior theory for over-parameterized noisy matrix sensing (2). Here $\widehat{X}$ denotes the recovered matrix, and $\sigma^2$ is the variance proxy for the additive noise.

| | RIP requirement | Initialization | Statistical error $\|\widehat{X} - X_\natural\|_F^2$ |
|---|---|---|---|
| [23, 24, 29] $(r_\natural \leq r \leq \tilde{\mathcal{O}}(m/n))$ | $2r$-RIP | Spectral initialization: $U_0 U_0^\top$ close to $X_\natural$ | $\tilde{\mathcal{O}}(\sigma^2 nr/m)$ |
| Ours $(r \geq r_\natural)$ | $2r_\natural$-RIP | Small random initialization | $\tilde{\mathcal{O}}(\sigma^2 nr_\natural/m)$ |

**Overview of our methods and contributions** This paper addresses the above issues, the over-fitting and the stronger requirement of $\mathcal{A}$, in the over-parameterized noisy matrix recovery problem (2). More precisely, by utilizing recent algorithm design and analysis advancements for over-parametrized matrix sensing [21, 25, 27, 28], particularly [25], we show that gradient descent (GD) with small random initialization (SRI) generates iterations toward the ground-truth matrix and achieves nearly information-theoretically optimal recovery within $\tilde{\mathcal{O}}(1)$ steps.

**Theorem 1.1** (Informal). *Suppose that $\mathcal{A}$ satisfies $2r_\natural$-RIP and consider gradient descent (GD) with small random initialization (SRI) and any $r \geq r_\natural$. Within the first $\tilde{\mathcal{O}}(1)$ steps, one of the iterates achieves statistical error of $\tilde{\mathcal{O}}(\sigma^2 nr_\natural/m)$.*

As summarized in Table 1, our work improves upon [23, 24, 29] by showing that gradient descent can achieve minimax optimal statistical error $\tilde{\mathcal{O}}(\sigma^2 nr_\natural/m)$ with only $2r_\natural$-RIP requirement on $\mathcal{A}$, even in the extreme over-parameterized case $r = n$.

However, GD with SRI alone is not practical. Because the method so far could not identify the iterate that achieves the minimax error and will eventually overfit the noise if not stopped properly (See Figure 1c). To this end, we introduce a practical and efficient stopping strategy for over-parametrized matrix sensing based on the classical validation set approach. In particular, when training our model, we hold out a subset $(y_{\text{val}}, \mathcal{A}_{\text{val}})$ from the data set. We then use the validation loss $\frac{1}{2}\left\|\mathcal{A}_{\text{val}}\left(U_t U_t^\top\right) - y_{\text{val}}\right\|_2^2$ to monitor the recovery error $\frac{1}{2}\|U_t U_t^\top - X_\natural\|_F^2$ (see red and black curves in Figure 1c) and detect the valley of the recovery error curve. We have the following guarantee.

**Theorem 1.2** (Informal). *The validation approach identifies an iterate with a statistical error of $\tilde{\mathcal{O}}(\sigma^2 nr_\natural/m)$, which is minimax optimal.*

This result shows that our detected iterate has a recovery error that is minimax optimal. Hence, the error is close to the best iterate, which is further verified by Figure 1c and experiments in Section 4. Henceforth, the two issues are now addressed theoretically and practically.

**Relationship to the noiseless case and the classical validation approach** We note that it is expected that the results from the noiseless case [21, 25] can be extended to the noisy case to a certain extent. What is surprising to us is that the extension is actually *statistically optimal*. Achieving this requires careful and delicate handling of the error caused by the noise and different techniques beyond simply bookkeeping of the noiseless case (see detailed explanation in Section 2.2 and Section C.3). As for stopping via a validation set, though it is a common practice, due to the hardness of trajectory analysis and nonconvexity, it usually lacks some theoretical guarantees. In this work, we show that this simple approach is provably effective, and in fact *optimal*, in the setting of overparametrized matrix sensing.

**Extension to image recovery with a deep image prior (DIP)** Learning over-parameterized models is becoming increasingly important in machine learning. Beyond low-rank matrix recovery problem, over-parameterized model has also been formally studied for several other important problems, including compressive sensing [31, 32] (which also uses a similar validation approach), logistic regression on linearly separated data [33], nonlinear least squares [34], deep linear neural networks and matrix factorization [27, 35–37], deep image prior [38, 39] and so on. Among these, the *deep image prior* (DIP) is closely related to the matrix recovery problem. It over-parameterizes an image

by a deep network, which is a non-linear multi-layer extension of the factorization $UU^\top$. While DIP has shown impressive results on image recovery tasks, it requires appropriate early stopping to avoid overfitting. The works [39, 40] propose an early stopping strategy by either training a coupled autoencoder or tracking the trend of the variance of the iterates, which are more complicated than the validation approach. In Section 4, we demonstrate by experiments that the proposed validation approach,

*partitioning the image pixels into a training set and a validation set,*

can be used to identify appropriate stopping for DIP efficiently. The novelty lies in partitioning the image pixels, traditionally considered one giant piece for training.

**Paper organization**  The rest of the paper is organized as follows. We conclude this introduction with a related work section to better position our work in the literature. We also include a notation paragraph explaining the notation used in this paper. In Section 2, we describe gradient descent with small random initialization in detail and the formal version of our first informal theorem, Theorem 2.3, that GD with SRI has a minimax optimal iterate. Next, in Section 3, we explain the validation approach for early stopping and the formal statement, Theorem 3.1, of our second informal theorem, that one can identify which iterate in GD with SRI is minimax optimal. After the theoretical analysis, in Section 4, we proceed with numerical verification of our theoretical results. We further extend our methodology to matrix completion and deep image prior, showing encouraging results. We summarize the paper in Section 5 and provide future directions.

**Notions and notations**  We denote the singular values of a matrix $X \in \mathbb{R}^{n \times n}$ by $\sigma_1(X) \geq \cdots \geq \sigma_n(X)$. The condition number of $X_\natural$ is denoted as $\kappa = \frac{\sigma_1(X_\natural)}{\sigma_{r_\natural}(X_\natural)}$. For any matrix $Z$, we use $\|Z\|, \|Z\|_\mathrm{F}$, $\|Z\|_*$ to denote its spectral norm (the largest singular value), the Frobenius norm (the $\ell_2$ norm of the singular values), and the nuclear norm (the sum of singular values). We equip the space $\mathbb{R}^m$ with the dot product and the space $\mathbb{R}^{n \times n}$ with the standard trace inner product. The corresponding norms are the $\ell_2$ norm $\|\cdot\|_2$ and $\|\cdot\|_F$ respectively. For a subspace $L \subset \mathbb{R}^n$, we use $V_L$ to denote an orthonormal representation of the subspace space of $L$, i.e., $V_L$ has $\dim(L)$ orthonormal columns and the columns span $L$. Moreover, we denote a representation of the orthogonal space of $L$ by $V_{L^\perp}$. We use the same notations $V_L$ and $V_{L^\perp}$ representing the column space of a matrix $L$ and its orthogonal complement, respectively. For two quantities $a, b \in \mathbb{R}$, the inequalities $a \gtrsim b$ and $b \lesssim a$ mean $b \leq ca$ for some universal constant $c$. A random variable $Z$ with mean $\mu$ is subgaussian with variance proxy $\sigma$, denoted as $Z \sim \mathrm{subG}(\sigma)$, if $\mathbb{E}(\exp(\lambda(Z - \mu))) \leq \exp(\sigma^2 \lambda^2 / 2)$ for any $\lambda \in \mathbb{R}$.

## 2. GD with SRI has a minimax optimal iterate

In this section, we first present gradient descent for Problem (2) and a few preliminaries. Next, we present the main result, which is that gradient descent coupled with small random initialization has an iterate achieving optimal statistical error.

### 2.1. Gradient descent and preliminaries

Gradient descent proceeds as follows: pick a stepsize $\eta > 0$, an initialization direction $U \in \mathbb{R}^{n \times r}$ and a size $\alpha > 0$,
$$\text{initialize } U_0 = \alpha U, \text{ and iterate } U_{t+1} = U_t - \eta \nabla f(U_t), \tag{3}$$
where $\nabla f(U_t) = \frac{1}{m} \cdot \left( \mathcal{A}^*(\mathcal{A}(U_t U_t^\top - X_\natural)) - \mathcal{A}^*(e) \right) U_t$. Here, $\mathcal{A}^*$ is the adjoint operator of $\mathcal{A}$.

To analyze the behavior of the gradient descent method (3), we consider the following restricted isometry condition [1], which states that the map $\mathcal{A}$ approximately preserves the $\ell_2$ norm between its input and output spaces.

**Definition 2.1.**  $[(k, \delta)$-RIP] *A linear map* $\mathcal{A} : \mathbb{R}^{n \times n} \to \mathbb{R}^m$ *satisfies* $(k, \delta)$ *restricted isometry property (RIP) for some integer* $k > 0$ *and* $\delta \in [0, 1]$ *if for any matrix* $X \in \mathbb{R}^{n \times n}$ *with* $\mathrm{rank}(X) \leq k$, *the inequalities* $(1 - \delta)\|X\|_F^2 \leq \frac{\|\mathcal{A}(X)\|_2^2}{m} \leq (1 + \delta)\|X\|_F^2$ *hold.*

Let $\mathcal{A}(Z) = (\langle A_1, Z \rangle, \ldots, \langle A_m, Z \rangle)^\top$ for any $Z \in \mathbb{R}^{n \times n}$. According to [2, Thereom 2.3], the $(k, \delta)$-RIP condition is satisfied with high probability if each sensing matrix $A_i$ contains iid subgaussian entries and $m \gtrsim \frac{nk \log(1/\delta)}{\delta^2}$.

Thus, if a linear map $\mathcal{A}$ satisfies RIP, then $\frac{1}{m} \|\mathcal{A}(X)\|_2^2$ is approximately equal to $\|X\|_F^2$ for $X$ with low rank. Additionally, the nearness in the function value actually implies the nearness in the gradient under certain norms, as the following proposition states.

**Proposition 2.2.** [25, Lemma 7.3] *Suppose $\mathcal{A}$ satisfies $(k, \delta)$-RIP. Let $\mathcal{I}$ denote the identity map on $\mathbb{R}^{d \times d}$. Then for any matrix $X \in \mathbb{R}^{n \times n}$ with rank no more than $k$, and any matrix $Z \in \mathbb{R}^{n \times n}$ we have the following two inequalities:*

$$\left\| \left( \mathcal{I} - \frac{\mathcal{A}^* \mathcal{A}}{m} \right)(X) \right\| \leq \delta \|X\|_F, \tag{4}$$

$$\left\| \left( \mathcal{I} - \frac{\mathcal{A}^* \mathcal{A}}{m} \right)(Z) \right\| \leq \delta \|Z\|_*. \tag{5}$$

To see the usefulness of the above bound, letting $\mathcal{D} = \mathcal{I} - \frac{\mathcal{A}^* \mathcal{A}}{m}$, we may decompose (3) as the following:

$$U_{t+1} = U_t - \eta(U_t U_t^\top - X_\natural) U_t + \eta[\mathcal{D}(U_t U_t^\top - X_\natural)] U_t. \tag{6}$$

Thus we may regard $[\mathcal{D}(U_t U_t^\top - X_\natural)] U_t$ as a perturbation term and focus on analyzing $U_{t+1} = U_t - \eta(U_t U_t^\top - X_\natural) U_t$. To control the perturbation, we use Proposition 2.2. We defer the detailed explanation of its usage to Section C.1.

## 2.2. Optimal statistical error guarantee

We now give a rigorous statement showing that there is a gradient descent iterate $U_t$, achieving the optimal statistical error.

**Theorem 2.3.** *We assume that $\mathcal{A}$ satisfies $(2r_\natural, \delta)$-RIP with $\delta \leq c\kappa^{-2} r_\natural^{-1/2}$. Suppose the noise vector $e$ has independent mean-zero $SubG(\sigma^2)$ entries and $m \gtrsim \kappa^2 n \frac{\sigma^2}{\sigma_{r_\natural}^2(X_\natural)}$. Further suppose the stepsize $\eta \leq c\kappa^{-2} \|X_\natural\|^{-1}$ and $r \geq r_\natural$. If the scale of initialization satisfies*

$$\alpha \lesssim \min \left\{ \frac{1}{\kappa^4 n^4} \left( C\kappa n^2 \right)^{-6\kappa} \sqrt{\|X_\natural\|}, \ \kappa \sqrt{\frac{n r_\natural}{m \|X_\natural\|}} \sigma \right\}.$$

*then after $\hat{t} \lesssim \frac{1}{\eta \sigma_{\min}(X_\natural)} \log \left( \frac{C n \kappa \sqrt{\|X_\natural\|}}{\alpha} \right)$ iterations, we have*

$$\|U_{\hat{t}} U_{\hat{t}}^\top - X_\natural\|_F^2 \lesssim \sigma^2 \kappa^2 \frac{r_\natural n}{m}.$$

*with probability at least $1 - \tilde{C}/n - \tilde{C} \exp(-\tilde{c} r)$, here $C, \tilde{C}, c, \tilde{c} > 0$ are fixed numerical constants.*

We note that unlike those bounds obtained in [23, 24], our final error bound $\mathcal{O}(\kappa^2 \sigma^2 \frac{n r_\natural}{m})$ has *no extra* logarithmic dependence on the dimension $n$ or the over-specified rank $r$. The initialization size $\alpha$ needs to be dimensionally small, which we think is mainly an artifact of the proof. We set $\alpha = 10^{-6}$ in our experiments. We explain the rationale behind this small random initialization in Section C.1. We note that the requirement that $\delta$ needs to be smaller than $\frac{1}{\sqrt{r_\natural}}$ might be relaxed further using the technique developed in [41] for some randomly generated $\mathcal{A}$. The condition $m \gtrsim \kappa^2 n \frac{\sigma^2}{\sigma_{r_\natural}^2(X_\natural)}$ ensures the noise matrix $\mathcal{A}^*(e)$ is small enough compared to the ground-truth $X_\natural$.

To prove Theorem 2.3, we utilize the following theorem (proved in Section C), which requires a deterministic condition on the noise matrix $\mathcal{A}^*(e)$ and allows a larger range for choosing $\alpha$.

**Theorem 2.4.** *Instate the same assumption in Theorem 2.3 for $\mathcal{A}$, the stepsize $\eta$, and the parametrized rank $r$. Let $E = \frac{1}{m} \mathcal{A}^*(e)$ and assume $\|E\| \leq c\kappa^{-1} \sigma_{\min}(X_\natural)$. If the scale of initialization satisfies*

$$\alpha \lesssim \frac{1}{\kappa^4 n^4} \left( C\kappa n^2 \right)^{-6\kappa} \sqrt{\|X_\natural\|},$$

*then after $\hat{t} \lesssim \frac{1}{\eta\sigma_{\min}(X_\natural)} \log\left(\frac{Cn\kappa\sqrt{\|X_\natural\|}}{\alpha}\right)$ iterations, we have*

$$\frac{\|U_{\hat{t}}U_{\hat{t}}^\top - X_\natural\|_{\mathrm{F}}}{\|X_\natural\|} \lesssim \frac{\alpha}{\sqrt{\|X_\natural\|}} + \sqrt{r_\natural}\kappa\frac{\|E\|}{\|X_\natural\|}$$

*with probability at least $1 - \tilde{C}/n - \tilde{C}\exp(-\tilde{c}r)$, here $C, \tilde{C}, c, \tilde{c} > 0$ are fixed numerical constants.*

**Proof strategy of Theorem 2.4** To prove Theorem 2.4, we first decompose the iterate $U_t$ into a signal matrix and an error matrix. We then analyze the first three phases of GD with SRI among its four phases: (i) subspace alignment: column space of the signal matrix gets closer to that of $X_\natural$, (ii) signal strength increasing: the strength of the signal matrix increases, (iii) local convergence up to statistical error: the distance square $\|U_tU_t^\top - X_\natural\|^2$ decreases up to $\mathcal{O}(\sigma^2 nr_\natural/m)$, and (iv) over-fitting phase: $\|U_tU_t^\top - X_\natural\|$ increases while $f(U_t)$ keeps shrinking. We defer the decomposition definition, further explanation of the proof, and the proof details to Section C. Our decomposition and characterization of the three phases closely follow the analysis in [25] with two critical adjustments: (1) we provide extra leeway in handling the *approximately* low-rank matrice $U_tU_t^\top - X_\natural$ (See Lemma C.3 and Remark C.4), and (2) we use the spectral norm in the analysis of the local convergence phase that allows us to use Proposition 2.2, while the norm used in [25] would require a more stringent condition which is unlikely to hold for the iterates $U_t$ (See the proof of Lemma C.2 and Footnote 7). These adjustments allow us to deal with the noise matrix $E$ and achieve *the optimal statistical error*.

Let us now prove Theorem 2.3.

*Proof of Theorem 2.3.* Consider the following bound on $\|\mathcal{A}^*(e)\|$:

$$\frac{1}{m}\|\mathcal{A}^*(e)\| \overset{(a)}{\lesssim} \sqrt{\frac{n}{m}}\sigma \overset{(b)}{\leq} c\kappa^{-1}\sigma_{r_\natural}(X_\natural).$$

The inequality $(a)$ comes from standard random matrix theory [30, Lemma 1.1] and the step $(b)$ is due to our assumption on $m$. Now Theorem 2.3 simply follows from Theorem 2.4 by plugging in the above bound and the choice of $\alpha$. $\qquad\square$

In Theorem 2.3, we only know that some iterate $U_{\hat{t}}$ will achieve the optimal statistical error, and the question of how to pick such iterate remains. We detail the procedure in Section 3.

## 3. Stopping via the validation approach

In this section, we propose an efficient method for stopping, i.e., finding the iterate $X_t = U_tU_t^\top$ that has nearly the smallest recovery error $\|X_t - X_\natural\|_F$.

**The validation approach** We exploit the *validation* approach, a common technique used in machine learning. Specifically, the data $(y, \mathcal{A})$ can be explicitly expressed as $\{(y_i, A_i)\}_{i=1}^m$ where each data sample $(y_i, A_i) \in \mathbb{R}^{1+n\times n}$ and $y_i = [\mathcal{A}(X_\natural)]_i = \mathrm{trace}(A_i^\top X_\natural)$. We randomly split the set $\{1, \ldots, m\}$ to get a partition, $\mathcal{I}_{\text{train}}$ and $\mathcal{I}_{\text{val}}$. We then split the data into $m_{\text{val}}$ validation samples with index set $\mathcal{I}_{\text{val}}, \{(y_i, A_i)\}_{i\in\mathcal{I}_{\text{val}}}$, and $m_{\text{train}}$ training samples with index set $\mathcal{I}_{\text{train}}, \{(y_i, A_i)\}_{i\in\mathcal{I}_{\text{train}}}$. We denote the measurement vector in the validation samples as $y_{\text{val}} \in \mathbb{R}^m$ where $[y_{\text{val}}]_i = 0$ if $i \notin \mathcal{I}_{\text{val}}$ and $[y_{\text{val}}]_i = y_i$ if $i \in \mathcal{I}_{\text{val}}$. In other words, we replace the entries in $y$ whose index is not in the validation set with 0. We define $e_{\text{val}}$ (the noise vector for the validation set) similarly. The linear operator for the validation set $\mathcal{A}_{\text{val}} : \mathbb{R}^{n\times n} \to \mathbb{R}^m$ is $[\mathcal{A}_{\text{val}}(X)]_i = 0$ if $i \notin \mathcal{I}_{\text{val}}$ and $\mathrm{trace}(A_iX)$ for $i \in \mathcal{I}_{\text{val}}$, for any $X \in \mathbb{R}^{n\times n}$. The vectors $y_{\text{train}}, e_{\text{train}}$, and the map $\mathcal{A}_{\text{train}}$ are defined similarly. The training loss $f_{\text{train}}$ is $f_{\text{train}} : U \mapsto \frac{1}{2m_{\text{train}}}\|\mathcal{A}_{\text{train}}(UU^\top) - y_{\text{train}}\|_2^2$. Next, we present the algorithm of the GD with SRI combined with the validation approach in Algorithm 1.

**Algorithm 1** GD with SRI combined with the validation approach

---

**Input:** data $\{(y_i, A_i)\}_{i=1}^m$, parametrized rank $r$, initial scale $\alpha$, stepsize $\eta$, and iteration number $T$

    **Step 1:** Split the data into a training set $\{(y_i, A_i)\}_{i \in \mathcal{I}_{\text{train}}}$ of size $m_{\text{train}}$ and a validation set $\{(y_i, A_i)\}_{i \in \mathcal{I}_{\text{val}}}$ of size $m_{\text{val}}$

    **Step 2:** Initialize $U_0 = \alpha U$ where $U \in \mathbb{R}^{n \times r}$ has iid standard Gaussian entries

    **Step 3:** for $t = 1, 2, \ldots, T$

            Compute $U_t = U_{t-1} - \eta \nabla f_{\text{train}}(U_{t-1})$ and the

            validation error $v_t = \frac{1}{m_{\text{val}}} \left\| \mathcal{A}_{\text{val}}(U_t U_t^\top) - y_{\text{val}} \right\|_2^2$.

            end for

**Output:** $U_{\hat{t}}$ where $\hat{t} = \arg\min_{1 \leq t \leq T} v_t$.

---

**Guarantee of Algorithm 1** We present our main theorem (proved in Section D) for Algorithm 1.

**Theorem 3.1.** *Consider Algorithm 1. Fix $\epsilon \in (0, 1)$. Instate the same assumption in Theorem 2.3 with $(y, \mathcal{A})$ replaced by $(y_{train}, \mathcal{A}_{train})$. Also suppose $(y_{train}, \mathcal{A}_{train})$ and $(y_{val}, \mathcal{A}_{val})$ are independent to each and that each entry of $A_i$, $i \in \mathcal{I}_{val}$, is iid subG$(c_1)$ with mean zero and variance 1 and each $e_i$, $i \in \mathcal{I}_{val}$ is iid subG$(c_2 \sigma^2)$ with mean zero. Finally suppose $\frac{(nr_\natural)^2 \kappa^2 m_{val}}{m_{train}^2} \geq C \frac{\log T}{\log(\frac{1}{\epsilon})}$ and $T > C \kappa^4 \log(\kappa(n + r))$ for some universal $C > 0$. Also let $\hat{t} = \arg\min_{1 \leq t \leq T} v_t$. Then, with probability at least $1 - \epsilon$, we have*

$$\|U_{\hat{t}} U_{\hat{t}}^\top - X_\natural\|_F^2 \leq C \frac{\kappa^2 \sigma^2 r_\natural n}{m_{train}}. \tag{7}$$

The condition of Theorem 3.1 is satisfied if (1) the sensing matrix $A_i$ and the noise vector $e$ have iid subgaussian entries, (2) the sample size $m_{\text{train}} \gtrsim \max\{nr_\natural^2 \kappa^4, \kappa^2 n \frac{\sigma^2}{\sigma_{r_\natural}^2(X_\natural)}\}$ and $T > C\kappa^4 \log(\kappa(n + r))$, and (3) the configuration of $m_{\text{val}}$, $m_{\text{train}}$, and $T$ satisfies $\frac{(nr_\natural)^2 \kappa^2 m_{\text{val}}}{m_{\text{train}}^2} \geq C \frac{\log T}{\log(\frac{1}{\epsilon})}$. The conditions (1) and (2) ensure the validity of Theorem 2.3 and are rather standard in the literature for matrix sensing [5, 8, 10]. The condition (3) is for the validity of the validation approach and holds for many practical scenarios for moderate condition numbers. For example, (3) holds if $m_{\text{val}}$ and $m_{\text{train}}$ are on the order of magnitude as $nr_\natural$, and $T = \mathcal{O}(\max\{n^2, \kappa^4 \log \kappa\})$. In general, the inequality holds if (a) the two quantities $m_{\text{val}}$ and $m_{\text{train}}$ are of the same order of magnitude and on the order of $o(n^2)$, and (b) the iteration number $T$ is polynomial but not exponential in $\kappa$ and $n$.

# 4. Numerics and Extensions

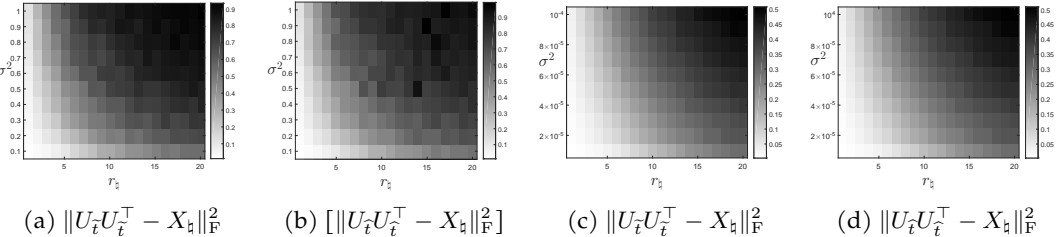

    (a) $\|U_{\tilde{t}} U_{\tilde{t}}^\top - X_\natural\|_F^2$     (b) $[\|U_{\hat{t}} U_{\hat{t}}^\top - X_\natural\|_F^2]$     (c) $\|U_{\tilde{t}} U_{\tilde{t}}^\top - X_\natural\|_F^2$     (d) $\|U_{\hat{t}} U_{\hat{t}}^\top - X_\natural\|_F^2$

Figure 2: Recovery error (whiter indicates better recovery) of Algorithm 1 for over-parameterized noisy matrix sensing (a & b) and matrix completion (c & d). (a) and (c) uses the iterate $U_{\tilde{t}}$ closest to $X_\natural$, while (b) and (d) uses the iterate $U_{\hat{t}}$ has the smallest validation loss.

In this section, we conduct a set of experiments on matrix sensing and matrix completion to demonstrate the performance of gradient descent with validation approach for over-parameterized matrix recovery. We also conduct experiments on image recovery with deep image prior to show potential applications of the validation approach for other related unsupervised learning problems.

**Matrix sensing** In these experiments, we generate a rank-$r_\natural$ matrix $X_\natural \in \mathbb{R}^{n \times n}$ by setting $X_\natural = U_\natural U_\natural^\top$ with each entries of $U_\natural \in \mathbb{R}^{n \times r_\natural}$ being normally distributed random variables, and then normalize $X_\natural$ such that $\|X_\natural\|_F = 1$. We then obtain $m$ measurements $y_i = \langle A_i, X_\natural \rangle + e_i$ for $i = 1, \ldots m$,

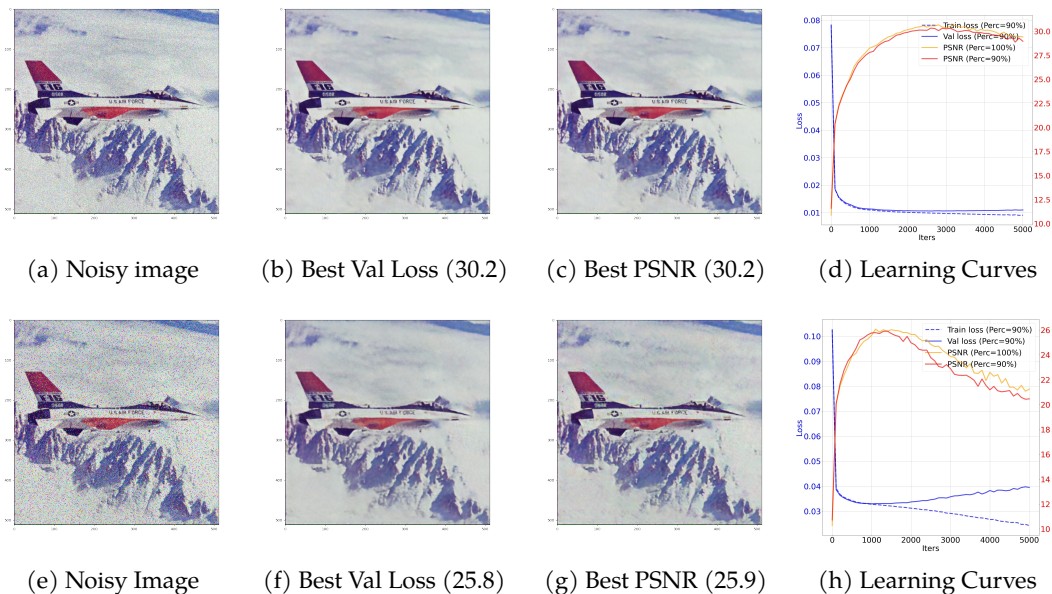

Figure 3: Image denoising by DIP. From left to right: the noisy image, the image with the smallest validation loss, the image with the best PSNR, and the learning curves in terms of training loss, validation loss, PSNR. Here, the top row is for Gaussian noise with mean 0 and variance 0.1, while bottom row is for salt and pepper noise where 10 percent of pixels are randomly corrupted. In (d) and (h), Perc = 90% means we train UNet on 90% of pixels and the rest of 10% pixels are used for validation; Perc = 100% means we train the network with the entire pixels. The model is optimized with Adam for 5000 iterations. The numbers listed in the captions show the corresponding PSNR.

where entries of $A_i$ are i.i.d. generated from standard normal distribution, and $e_i$ is a Gaussian random variable with zero mean and variance $\sigma^2$. We set $n = 50, m = 1000$, and vary the rank $r_\natural$ from 1 to 20 and the noise variance $\sigma^2$ from 0.1 to 1. We then split the $m$ measurements into $m_{\text{train}} = 900$ training data and $m_{\text{val}} = 100$ validation data to get the training dataset $(y_{\text{train}}, \mathcal{A}_{\text{train}})$ and validation dataset $(y_{\text{val}}, \mathcal{A}_{\text{val}})$, respectively. To recover $X_\natural$, we use Algorithm 1 with $r = n, \eta = 0.5, T = 500$, and $\alpha = 10^{-6}$, which generates iterates $U_t$. Within all the generated iterates $U_t$, we select

$$\widetilde{t} = \arg\min_{1 \le t \le T} \|U_t U_t^\top - X_\natural\|_{\text{F}}, \quad \widehat{t} = \arg\min_{1 \le t \le T} \|\mathcal{A}_{\text{val}}(U_t U_t^\top) - y_{\text{val}}\|_2. \tag{8}$$

That is, the iterate $U_{\widetilde{t}}$ is the closest to $X_\natural$, while $U_{\widehat{t}}$ achieves the smallest validation loss.

For each pair of $r_\natural$ and $\sigma^2$, we conduct 10 Monte Carlo trails and compute the average of the recovered errors for $\|U_{\widetilde{t}} U_{\widetilde{t}}^\top - X_\natural\|_{\text{F}}^2$ and $\|U_{\widehat{t}} U_{\widehat{t}}^\top - X_\natural\|_{\text{F}}^2$. The results are shown in Figure 2 (a) and (b). In both plots, we observe that the recovery error is proportional to $r_\natural$ and $\sigma^2$, consistent with Theorem 2.3 and Theorem 3.1. Moreover, comparing Figure 2 (a) and (b), we observe similar recovery errors for both $U_{\widetilde{t}}$ and $U_{\widehat{t}}$, demonstrating the performance of the validation approach.

**Matrix completion** In the second set of experiments, we test the performance on matrix completion, where we want to recover a low-rank matrix from incomplete measurements. Similar to the setup for matrix sensing, we generate a rank-$r_\natural$ matrix $X_\natural$ and then randomly obtain $m$ entries with additive Gaussian noise of zero mean and variance $\sigma^2$. In this case, each selected entry can be viewed as obtained by a sensing matrix that contains zero entries except for the location of the selected entry being one. Thus, we can apply the same gradient descent to solve the over-parameterized problem to recover the ground-truth matrix $X_\natural$. In these experiments, we set $n = 50, m = 1000, \eta = 0.5, T = 500, \alpha = 10^{-3}$, and vary the rank $r_\natural$ from 1 to 20 and the noise variance $\sigma^2$ from $10^{-5}$ to $10^{-4}$. We display the recovered error in Figure 2 (c) and (d). Similar to the results in matrix sensing, we also observe from Figure 2 (c) and (d) that gradient descent with the validation approach produces a good solution for solving over-parameterized noisy matrix completion problem.

**Deep image prior** In the last set of experiments, we test the performance of the validation approach on image restoration with deep image prior (DIP). We follow the experiment setting as the work [38, 42] for the denoising task, except that we randomly hold out 10 percent of pixels to decide the images with the best performance during the training procedure. Concretely, we train the identical UNet network on the normalized standard dataset[6] with different types of optimizers, noises, and losses. In particular, the images are corrupted by two types of noises: $(i)$ Gaussian noise with mean 0 and standard deviation from 0.1 to 0.3, and $(ii)$ salt and pepper noise where a certain ratio (between 10% to 50%) of randomly chosen pixels are replaced with either 1 or 0. We use SGD with a learning rate 5 and Adam with a learning rate 0.05. We evaluate the PSNR, a common measure in DIP, the higher the better, and the validation loss (Val loss) on the hold-out pixels across all experiment settings.

In Figure 3, we use Adam to train the network with the L2 loss function for the Gaussian noise (at the top row) and the salt and pepper noise (at the bottom row) for 5000 iterations. From left to right, we plot the noisy image, the image with the smallest validation loss, the image with the best PSNR w.r.t. ground-truth image, the learning curves, and the dynamic of training progress. We observe that for the case with Gaussian noise in the top row, our validation approach finds an image with PSNR 30.1544, which is *close to the best PSNR* 30.2032 through the entire training progress. A similar phenomenons also appear for salt and pepper noise, for which our validation approach finds an image with a PSNR of 25.7735, which is *close to the best PSNR* 25.8994. We note that finding the image with the best PSNR is impractical without knowing the ground-truth image. On the other hand, as shown in the learning curves, the network overfitts the noise without early stopping. Finally, one may wonder without the holding out 10 percent of pixels will decreases the performance. To answer this question, we also plot the learning curves (orange lines) in Figure 3 (d) and (h) for training the UNet with entire pixels. We observe that the PSNR will only be improved slightly by using the entire image, achieving PSNR 30.6762 for Gaussian noise and PSNR 26.0728 for salt-and-pepper noise. Moreover, we further test the validation approach across different images, loss functions, and noise levels in the supplement and observe its success in all those settings.

## 5. Discussion

On the matrix recovery side, we analyzed gradient descent for the noisy over-parameterized matrix recovery problem where the rank is overspecified, and the global solutions overfit the noise and do not correspond to the ground-truth matrix. Under the restricted isometry property for the measurement operator, we showed that `GD` with `SRI` stopped by a validation method achieves nearly statistically optimal recovery.

Based on recent works in the noiseless regime [28, 43], we think extending our analysis to a nonsymmetric matrix $X_\natural$, for which we need to optimize over two factors $U$ and $V$, is promising, while extending our analysis to the matrix completion problem with Bernoulli sampling, as supported by the experiments, is still challenging due to the dependence between the training and validation samples. Moreover, it would be interesting to see whether one can actually stop the method early, e.g., the method stops after the validation error keeps increasing for 10 consecutive iterations, to have some computational savings. Answering this question requires the characterization of the overfitting phase, which our current analysis does not cover. We believe extending our analysis to the above settings could provide deeper insights into the current analysis of overparametrization and expand the applicability of the methods described here.

On the image recovery side, we extended our validation approach, by partitioning the pixels of the image into training and validation sets, to deep images prior to image recovery and achieved encouraging numerical results. It would be interesting to see whether our validation approach could help other networks for image recovery and general inverse problems, e.g., [44].

---

[6]http://www.cs.tut.fi/~foi/GCF-BM3D/index.htm#ref_results

# Acknowledgement

This work was supported by NSF grants CCF-2023166, CCF-2241298, ECCS-2409701, and IIS-2402952.

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

# Appendices

## A. Related work

Compared to the relatively mature literature of exact-parameterized case $r = r_\natural$ and the convex optimization approach to matrix sensing, the literature on overparametrized matrix sensing is more vibrant and proliferating. We refer the reader to [19, Section 5] and [5] for an overview of exact parametrization, and to [19, Section 4] and [45, Chapter 9] for comprehensive summaries of the convex approaches. In the following, we divide the discussion according to the noise level: the noiseless regime ($\sigma = 0$) and the dense noise regime ($\sigma > 0$). Additional discussion on related work on sparse and additive noise, not modeled in this paper, can be found at the end of this section.

Table 2: Summary of the related work and the position of this work.

|  | small random initialization | spectral initialization |
| --- | --- | --- |
| Noiseless regime ($\sigma = 0$) | [21, 25, 26, 28] | [23, 24, 29] |
| Noisy regime ($\sigma > 0$) | This work | [23, 24, 29] |

**Noiseless regime** In the noiseless regime, the work [21, 25, 26, 28] shows that among the first few iterates of gradient descent (GD) with small random initialization(SRI) for (2), there is one close to the ground-truth matrix $X_\natural$, where the closeness can be made arbitrarily small by changing the initialization scale. More precisely, [21] deals with the case $r = n$, while [25] deals with the general case $r > r_\natural$, and [28] further extends the analysis to the ground-truth matrix that is not necessarily PSD. In [26], the authors deal with the issue caused by the condition number of $X_\natural$. In these works, the operator $\mathcal{A}$ is only required to satisfy $2r_\natural$-RIP. The principal idea behind [21, 25, 26, 28] is that even though there exists an infinite number of solutions $U$ such that $\mathcal{A}(UU^\top) = y = \mathcal{A}(X_\natural)$, GD with SRI produces an *implicit bias* towards low-rank solutions [46]. Nevertheless, such a conclusion cannot be directly applied to the noisy case since a $U$ can overfit the noise and does not produce the desired solution $X_\natural$. As shown in Figure 1c, while the training loss (blue curve) keeps decreasing, the iterates $U_t$ eventually tend to overfit as indicated by the recovery error (black curve). By providing a practical stopping rule and extending the analysis to the noisy regime, our work shows that GD with SRI can find a minimax optimal estimator when stopped using the validation approach.

**Noisy regime**  In the noisy regime, the works [23, 24, 29] follow the more classical initialization method, the so-called *spectral initialization*, and then use the gradient descent method or its variants. As discussed earlier, the guarantees reflect the overfitting of the algorithm and require $2r$-RIP of the operator $\mathcal{A}$. It is worth noting that even in the noiseless regime, these algorithms require $2r$-RIP condition on $\mathcal{A}$. The later work [24, 29] elegantly resolves the issue of the slower convergence of vanilla gradient descent in the overparametrization regime when the number of iterations approaches infinity, observed in [23, 24, 47, 48], and theoretically confirmed in [49]. However, in the noisy regime, such quick convergence of methods in [24, 29] means overfitting quickly, though it is mild when $r$ is larger than but close to $r_\natural$.

**Summary and positioning of the current paper**  We summarize the prior discussion in Table 2. In short, our work follows the small initialization strategy used primarily in the noiseless regime and fills the missing piece in the noisy regime. Furthermore, as mentioned earlier, it provides a practical way to find an estimator that provably achieves minimax error in a logarithmic number of steps.

**Sparse additive noise**  Apart from dense Subgaussian noise discussed in the main text, over-parameterized matrix recovery has also been studied in the presence of *sparse* additive noise [42, 47, 50, 51], which either models the noise as an explicit term in the optimization objective [42], or uses subgradient methods with diminishing step sizes for the $\ell_1$ loss of [47, 50, 51]. We note the work [51] remarkably extends the approach of $\ell_1$ loss to the *dense* Gaussian noise case, based on the property called sign-RIP for the sensing operator. Unlike the standard RIP, the sign-RIP is only known to be satisfied by Gaussian distribution of $\mathcal{A}$, and may not hold for other sub-Gaussian distributions, such as the Rademacher distribution, as the rotation invariance of Gaussian is central to its proof. Also, the subgradient method requires diminishing step sizes, which need to be fine-tuned in practice.

# B.  Details of experiments in Section 1

In this section, we describe the detail of the experiments performed in Section 1.

**Experiment detail in Figure 1a**  We set $n = 50$, $m = 1000$, $\sigma = 0.3$, and $r_\natural = 5$. For the method proposed in [23], and each parametrized rank $r$ in $\{1, 5, 10, 15, 20, 25, 30, 35, 40, 45, 50\}$, we generate a new operator $\mathcal{A}$ with standard Gaussian entries, and a new $X_\natural = UU^\top/\|UU^\top\|_F$ where $U \in \mathbb{R}^{n \times r_\natural}$ with standard Gaussian entries. We run the gradient descent method coupled with spectral initialization in [23] (with a learning rate $\eta = 0.25$) for one thousand iterations and record the recovery error and training error of the last iterate $U_{1000}$. For each $r$, we repeat the process for 20 times, and average the recovery error and the training error.

For our method, and each parametrized rank $r$ in $\{1, 5, 10, 15, 20, 25, 30, 35, 40, 45, 50\}$, we again generate a new operator $\mathcal{A}$ with standard Gaussian entries, and a new $X_\natural = UU^\top/\|UU^\top\|_F$ where $U \in \mathbb{R}^{n \times r_\natural}$ with standard Gaussian entries. We use $0.95m$ samples for training, and $0.05m$ for validation. We run our method (Algorithm 1 with an initialization scale $\alpha = 10^{-9}$ and the above training and validation set (with a learning rate $\eta = 0.25$) for one thousand iteration, and record the recovery error of the iterate determined by the validation approach. For each $r$, we repeat the process for 20 times, and average the recovery error.

**Experiment detail in Figure 1b**  We set $m = 4nr_\natural$, $r_\natural = 5$, and $\sigma = 1/n$, and $r = n$. For each method and each choice of $n \in \{10, 20, 30, 40, 50, 60, 70, 80\}$, we repeat the process of generation of $\mathcal{A}$ and the matrix $X_\natural$, and the methods for 20 times. Each methods is performed with the same choices of parameter in Figure 1a.

**Experiment detail in Figure 1c**  We set $n = 50$, $m = 1200$, and $r_\natural = 5$. We generate an operator $\mathcal{A}$ with standard Gaussian entries, and a $X_\natural = UU^\top/\|UU^\top\|_F$ where $U \in \mathbb{R}^{n \times r_\natural}$ with standard Gaussian entries. We use $0.9m$ samples for training, and $0.1m$ for validation. We run our method (Algorithm 1 with an initialization scale $\alpha = 10^{-6}$ and the above training and validation set (with

a learning rate $\eta = 0.25$) for one thousand iteration, and record the recovery error, validation error, and the training loss.

# C. Proof of Theorem 2.4

In this section, we prove our main result, Theorem 2.4. We first present some preliminaries and shorthand notations. We then present the proof strategy. Next, we provide detailed lemmas to characterize the three phases described in the strategy. Theorem 2.4 follows immediately from these lemmas.

**Some preliminaries and shorthand notations** For the ease of the presentation, in the following sections, we absorb the additional $\frac{1}{\sqrt{m}}$ factor into $\mathcal{A}$ and that the linear map $\mathcal{A}$ satisfies $(k, \delta)$-RIP means that

$$\frac{\|\mathcal{A}(X)\|_2^2}{\|X\|_F^2} \in [1 - \delta, 1 + \delta], \quad \forall X \text{ with } \text{rank}(X) \le k.$$

We also decompose $X_\natural = U_\natural U_\natural^\top$. We write $\kappa_f = \frac{\sigma_1(U_\natural)}{\sigma_{r_\natural}(U_\natural)}$. Note that $\kappa_f^2 = \kappa$. We define the noise matrix $E = \mathcal{A}^*(e)$. We denote the product iterate $X_t = U_t U_t^\top$ and its difference to $X_\natural$ as $\Delta_t = X_\natural - U_t U_t^\top$. We shall define $W_t \in \mathbb{R}^{r \times r_\natural}$ and its orthogonal complement matrix $W_{t,\perp} \in \mathbb{R}^{r \times (r-r_\natural)}$ shortly. With $W_t$ and $W_{t,\perp}$ in mind, we denote the adjusted iterate, $\tilde{U}_t = U_t W_t$, its product iterate $\tilde{X}_t = \tilde{U}_t \tilde{U}_t^\top$, the orthogonal complement of the product $X_{t,\perp} = U_t W_{t,\perp} W_{t,\perp}^\top U_t^T$, and the difference of the product to $X_\natural$, $\tilde{\Delta}_t = X_\natural - U_t W_t W_t^\top U_t^T$.

## C.1. Proof strategy

Our proof is based on [25], which deals with the case $E = 0$, with a careful adjustment in handling the extra error caused by the noise matrix $E$. In this section, we outline the proof strategy and explain our contribution in dealing with the issues of the presence of noise. The main strategy is to show a signal term converges to $X_\natural$ up to some error while a certain error term stays small. To make the above precise, we present the following decomposition of $U_t$.

**Decomposition of** $U_t$ Consider the matrix $V_{X_\natural}^\top U_t \in \mathbb{R}^{r_\natural \times r}$ and its singular value decomposition $V_{X_\natural}^\top U_t = V_t \Sigma_t W_t^\top$ with $W_t \in \mathbb{R}^{r \times r_\natural}$. Also denote $W_{t,\perp} \in \mathbb{R}^{r \times (r-r_\natural)}$ as a matrix with orthnormal columns and is orthogonal to $W_t$. Then we may decompose $U_t$ into "signal term" and "error term":

$$U_t = \underbrace{U_t W_t W_t^\top}_{\text{signal term}} + \underbrace{U_t W_{t,\perp} W_{t,\perp}^\top}_{\text{error term}}. \tag{9}$$

The above decomposition is introduced in [25, Section 5.1] and may look technical at first sight as it involves singular value decomposition. A perhaps more natural way of decomposing $U_t$ is to split it according to the column space of the ground truth $X_\natural$ as done in [23, 47]: $U_t = V_{X_\natural} V_{X_\natural}^\top U_t + V_{X_\natural^\perp} V_{X_\natural^\perp}^\top U_t$. However, as we observed from the experiments (not shown here), with the small random initialization, though the signal term $V_{X_\natural}^\top U_t$ does increase at a fast speed, the error term $V_{X_\natural^\perp}^\top U_t$ could also increase and does not stay very small. Thus, with $2r_\natural$-RIP only, the analysis becomes difficult as $V_{X_\natural^\perp}^\top U_t$ could be potentially high rank and large in the nuclear norm.

**Critical quantities for the analysis** What kind of quantities shall we look at to analyze the convergence of $U_t U_t^\top$ to $X_\natural$? The most natural one perhaps is the distance measured by the Frobenius norm: $\|U_t U_t^\top - X_\natural\|_F$. However, this quantity is almost stagnant in the initial stage of the gradient descent dynamic with small random initialization (3). As in [25], we further consider the following three quantities to enhance the analysis: (a) the magnitude of signal term, $\sigma_{r_\natural}(U_t W_t)$, (b) the magnitude of error term, $\|U_t W_{t,\perp}\|$, and (c) the alignment of column space between signal to ground truth, $\|V_{X_\natural^\perp}^\top V_{U_t W_t}\|$. Here, we assume $U_t W_t$ is full rank, which can be ensured at initial random initialization and proved by induction for the remaining iterates. Note that by definition, we have the

following equality [25, Lemma 5.1], which is employed often in the analysis,

$$V_{X_\natural} U_t = V_{X_\natural} U_t W_t. \tag{10}$$

**Four phases of the gradient descent dynamic** (3)  Here we describe the four phases of the evolution of (3) and how the corresponding quantities change. The first three phases will be rigorously proved in the appendices.

1. The first phase is called the alignment phase. In this stage, there is an alignment between column spaces of the signal $U_t W_t$ and the ground truth $X_\natural$, i.e., the quantity $\|V_{X_\natural^\perp}^\top V_{U_t W_t}\|$ decreases and becomes small. Moreover, the signal $\sigma_{r_\natural}(U_t W_t)$ is larger than the error $\|U_t W_{t,\perp}\|$ at the end of the phase though both terms are still as small as initialization.

2. The second phase is the signal-increasing stage. The signal term $U_t W_t$ matches the scaling of $(X_\natural)^{\frac{1}{2}}$ (i.e., $\sigma_{r_\natural}(U_t W_t) \geq \frac{\sqrt{\sigma_{r_\natural}(X_\natural)}}{10}$) at a geometric speed, while the error term $\|U_t W_{t,\perp}\|$ is almost as small as initialization and the column spaces of the signal $U_t W_t$ and the ground truth $X_\natural$ still align with each other.

3. The third phase is the local convergence phase. In this phase, the distance $\|U_t U_t^\top - X_\natural\|_F$ starts geometrically decreasing up to the statistical error. The analysis of this stage deviates from the one in [25] due to the presence of noise. In this stage, both $\|U_t W_{t,\perp}\|$ and $\|V_{X_\natural^\perp}^\top V_{U_t W_t}\|$ are of similar magnitude as before.

4. The last phase is the over-fitting phase. Due to the presence of noise, the gradient descent method will fit the noise, and thus $\|U_t U_t^\top - X_\natural\|_F$ will increase, and $U_t$ approaches an optimal solution of (2) which results in over-fitting.

In short, we observe that the first two phases behave similarly to the noiseless case, and thus, we only provide the necessary details in adapting the proof in [25]. However, the third phase requires additional efforts to deal with the noise matrix $E$. Next, we describe the effect of small random initialization and why $2r_\natural$-RIP is sufficient.

**Blessing and curse of small random initialization**  As mentioned after Theorem 2.3, we require the initial size to be very small. Since the error term increases at a much lower speed compared with the signal term, small initialization ensures that $U_t W_t W_t^\top U_t$ gets closer to $X_\natural$ while the error $U_t W_{t,\perp}$ stays very small. The smallness of the error in the later stage of the algorithm is a blessing of small initialization. However, since $\alpha$ is very small and the direction $U$ is random, the signal $U_t W_t$ initially is also very weak compared to the size of $X_\natural$. The initial weak signal is a curse of small random initialization.

**Why is $2r_\natural$-RIP enough?**  Since we are handling $n \times r$ matrices, it is puzzling why, in the end, we only need $2r_\natural$ RIP of the map $\mathcal{A}$. As it shall become clear in the proof, the need for RIP is mainly for bounding $\|(\mathcal{I} - \frac{\mathcal{A}^*\mathcal{A}}{m})\Delta_t\|$. With the decomposition (9), we have

$$\|(\mathcal{I} - \mathcal{A}^*\mathcal{A})\Delta_t\| \leq \|(\mathcal{I} - \mathcal{A}^*\mathcal{A})(\tilde{\Delta}_t + X_{t,\perp})\| \overset{(a)}{\leq} \delta\|\tilde{\Delta}_t\| + \delta\|X_{t,\perp}\|_*. \tag{11}$$

Here, in the inequality $(a)$, we use the spectral-to-Frobenius bound (4) for the first term and the spectral-to-nuclear bound (5) for the second term. Recall that (i) $X_{t,\perp} = U_t W_{t,\perp}(U_t W_{t,\perp})^\top$ and the error term $U_t W_{t,\perp}$ is very small due to the choice of $\alpha$, and (ii) $\tilde{\Delta}_t$ is the quantity of interest. Thus, bounding $\|(\mathcal{I} - \mathcal{A}^*\mathcal{A})\Delta_t\|$ becomes feasible.

## C.2.  Analyzing the three phases and the proof of Theorem 2.3

We first show the lemma stating the progress after the first two phases. It rigorously characterizes the behavior of the three quantities $\sigma_{r_\natural}(U_t W_t)$, $\|V_{X_\natural^\perp}^\top V_{U_t W_t}\|$, and $\|U_t W_{t,\perp}\|$ at the end of the second phase.

**Lemma C.1.** *Let $U \in \mathbb{R}^{n \times r}$ be a random matrix with i.i.d. entries with distribution $\mathcal{N}\left(0, 1/\sqrt{r}\right)$. Assume that the linear map $\mathcal{A}$ satisfies $(2r_\natural, \delta)$ RIP with $\delta \leq \frac{c_1}{\kappa_f^4 \sqrt{r_\natural}}$ and the bound $\|E\| \leq c_1 \kappa_f^{-2} \sigma_{r_\natural}(X_\natural)$. Let $U_0 = \alpha U$ for any*

$$\alpha \leq c_1 \min \left\{ \frac{(C\kappa n^2)^{-6\kappa}}{\kappa^4 n^4} \sqrt{\|X_\natural\|}, \ \kappa \sqrt{\frac{n r_\natural}{m \|X_\natural\|}} \sigma \right\}. \tag{12}$$

*Assume that the step size satisfies $\eta \leq c_2 \kappa_f^{-2} \|U_\natural\|^2$. Then with probability at least $1 - C\exp(-cr) - \frac{C}{n}$, after at most $t_1$ iterations where $t_1 \lesssim \frac{1}{\eta \sigma_{\min}^2(U_\natural)} \left( \ln\left(C\kappa n^2\right) + \ln\left(\frac{\sigma_{\min}(U_\natural)}{\gamma}\right) \right)$ for some $\gamma \in \alpha[\frac{c}{n}, C\kappa^2 n^2]$, we have*

$$\sigma_{\min}\left(V_{X_\natural}^T U_t\right) \geq \frac{\sigma_{\min}(U_\natural)}{\sqrt{10}} \tag{13}$$

$$\|U_t W_{t,\perp}\| \leq 2\sigma_{\min}^{\frac{1}{8}}(U_\natural)\gamma^{\frac{7}{8}}, \tag{14}$$

$$\|U_t\| \leq 3\|U_\natural\|, \quad \text{and} \quad \|V_{X_\natural^\perp}^T V_{U_t W_t}\| \leq c_2 \kappa_f^{-2}. \tag{15}$$

*Here $c, C, c_1, c_2 > 0$ are absolute numerical constants.*

*Proof.* The proof can be adapted from the one in [25] dealing with the first two phases. Here, we only provide the necessary details:

1. In the first phase, one uses the proof of [25, Lemma 8.7] and replaces the iterated matrix $M = \mathcal{A}^*\mathcal{A}(X_\natural)$ with $M = \mathcal{A}^*\mathcal{A}(X_\natural) + E$.

2. In the second phase, one uses [25, Proof of Theorem 9.6, Phase II] and replaces the iterated matrix $(\mathcal{I} - \mathcal{A}^*\mathcal{A})(\Delta_t)$ by $(\mathcal{I} - \mathcal{A}^*\mathcal{A})(\Delta_t) + E$.

3. In combining the above proof, set the quantities $\epsilon$ and $\beta$ in [25, Lemma 8.7] to be $\frac{1}{n}$ and $\frac{\alpha}{4\gamma}$ respectively.

4. All the argument there then works for the noisy case with the extra condition: $\|E\| \leq \frac{c_1 \sigma_{r_\natural}(X_\natural)}{\kappa^2}$ for some small $c_1$.

$\square$

Different from the noiseless case, the iterate $U_t$ can only get close to $U_\natural$ up to some level due to the presence of noise. The following lemma characterizing the quantity $\|\Delta_t\|_F$ is our main technical endeavor, whose proof is in Section C.3.

**Lemma C.2.** *Instate the assumptions and notations in Lemma C.1. and define*

$$t_\Delta = \frac{\ln\left(\max\left\{1; \frac{\kappa_f}{\min\{r;n\} - r_\natural}\right\} \frac{\|U_\natural\|}{\gamma}\right)}{\eta \sigma_{\min}(U_\natural)^2}. \tag{16}$$

*If $r > r_\natural$, then after $\hat{t} - t_1 \lesssim t_\Delta$ iterations, it holds that*

$$\|\Delta_{\hat{t}}\|_F \lesssim \frac{(\min\{r;n\} - r_\natural)^{3/4} r_\natural^{1/2}}{\kappa_f^{3/16}} \cdot \gamma^{21/16} \|U_\natural\|^{21/16} + r_\natural^{1/2} \kappa_f^2 \|E\|.$$

*If $r = r_\natural$, then for any $t \geq t_1$, we have*

$$\|\Delta_t\|_F \lesssim r_\natural^{1/2}\left(1 - \frac{\eta}{400}\sigma_{\min}(U_\natural)^2\right)^{t - t_1} \|U_\natural\|^2 + r_\natural^{1/2}\|E\|\kappa_f^2.$$

We end the subsection with the proof of Theorem 2.4

*Proof of Theorem 2.4.* Theorem 2.4 is immediate by using C.1 and C.2 and the range of $\gamma$ and $\alpha$. $\square$

### C.3. Proof of Lemma C.2

We start with a lemma showing that $U_t U_t^T$ converges towards $X_\natural$ up to some statistical error when projected onto the column space of $X_\natural$. The proof of the lemma can be found in Section C.4.

**Lemma C.3.** *Assume that* $\|U_t\| \leq 3\|U_\natural\|$, $\|E\| \leq \|U_\natural\|^2$, $\|X_{t,\perp}\|_* \leq \|U_\natural\|^2$, *and* $\sigma_{\min}\left(\tilde{U}_t\right) \geq \frac{1}{\sqrt{10}}\sigma_{\min}(U_\natural)$. *Moreover, assume that* $\eta \leq c\kappa_f^{-2}\|U_\natural\|^{-2}$, $\|V_{X_\natural^\perp}^T V_{\tilde{U}_t}\| \leq c\kappa_f^{-2}$, *and*

$$\|\left(\mathcal{I} - \mathcal{A}^*\mathcal{A}\right)(\Delta_t)\| \leq c\kappa_f^{-2}\left(\|\tilde{\Delta}_t\| + \|X_{t,\perp}\|_*\right) \tag{17}$$

*where the constant* $c > 0$ *is chosen small enough. Then, it holds that*

$$\|V_{X_\natural}^T\Delta_{t+1}\| \leq \left(1 - \frac{\eta}{200}\sigma_{\min}^2(U_\natural)\right)\|V_{X_\natural}^T\Delta_t\| \quad +\eta\frac{\sigma_{\min}^2(U_\natural)}{100}\|X_{t,\perp}\|_* + 18\eta\|U_\natural\|^2\|E\|.$$

**Remark C.4.** *We note that an analogous result of Lemma C.3 is established in* [25, *Lemma 9.5*]. *Unfortunately, the condition there requires the RHS of* (17) *to be* $\|X_\natural - U_t U_t^\top\|$ *instead of* $\|X_\natural - U_t W_t W_t^\top U_t^T\| + \|U_t W_{t,\perp} W_{t,\perp}^\top U_t\|_*$. *Such a condition,* $\|\left(\mathcal{I} - \mathcal{A}^*\mathcal{A}\right)(\Delta_t)\| \leq c\kappa_f^{-2}\|X_\natural - U_t U_t^\top\|$, *can not be satisfied by* $2r_\natural$-*RIP alone as* $U_t U_t^\top$ *is not necessarily low rank. Indeed, in the later version* [52], *the author changes the condition in the lemma, though still different from the one here.*

Let us now prove Lemma C.2.

*Proof of Lemma C.2.* **Case** $r > r_\natural$**:** Set $\hat{t} := t_1 + t_{\tilde{\Delta}}$, where $t_{\tilde{\Delta}} = \left\lfloor \frac{300}{\eta\sigma_{\min}(U_\natural)^2} \ln\left(\kappa_f^{1/4}\frac{1}{16(\min\{r;n\}-r_\natural)}\frac{\|U_\natural\|^{7/4}}{\gamma^{7/4}}\right)\right\rfloor$. Note that $t_{\tilde{\Delta}} \lesssim t_\Delta$ from the range of $\gamma$. Denote $\xi_1 = 1 - \frac{\eta}{400}\sigma_{\min}^2(U_\natural)$, $\xi_2 = 1 + 80\eta c_2\sigma_{\min}^2(U_\natural)$, and $\xi_3 = 1 - \frac{\eta\sigma_{\min}^2(U_\natural)}{200}$. We first state our induction hypothesis for $t_1 \leq t \leq \hat{t}$:

$$\sigma_{\min}(U_t W_t) \geq \sigma_{\min}\left(V_{X_\natural}^T U_t\right) \geq \frac{\sigma_{\min}(U_\natural)}{\sqrt{10}}, \tag{18}$$

$$\|U_t W_{t,\perp}\| \leq \xi_2^{t-t_1}\|U_{t_1}W_{t_1,\perp}\|, \tag{19}$$

$$\|U_t\| \leq 3\|U_\natural\|, \tag{20}$$

$$\|V_{X_\natural^\perp}^T V_{U_t W_t}\| \leq c_2\kappa_f^{-2}, \tag{21}$$

$$\|V_{X_\natural}^T\Delta_t\| \leq 10\xi_1^{t-t_1}\|U_\natural\|^2 + 18\eta\|U_\natural\|^2\|E\|\sum_{\tau=t_1+1}^{t}\xi_3^{\tau-t_1-1}. \tag{22}$$

For $t = t_1$ we note the inequalities (18), (20), and (21) follow from Lemma C.1.[7] The inequality (19) trivially holds for $t = t_1$. For $t = t_1$, the inequality (22) holds due to the following derivation:

$$\|V_{X_\natural}^T\Delta_t\| = \|V_{X_\natural}^T\tilde{\Delta}_t\| \leq \|X_\natural\| + \|\tilde{X}_t\| \leq 10\|U_\natural\|^2.$$

The last step is due to $\|U_{t_1}W_{t_1}\| \leq \|U_{t_1}\| \leq 3\|U_\natural\|$ by (20).

Using triangle inequality, the bound for $\|\left(\mathcal{A}^*\mathcal{A} - \mathcal{I}\right)(\Delta_t)\|$ in [25, pp. 27 of the supplement], and the assumption on $E$, we see that $\|\left(\mathcal{A}^*\mathcal{A} - \mathcal{I}\right)(\Delta_t) + E\| \leq 40c_1\kappa_f^{-2}\sigma_{r_\natural}(U_\natural)^2$. This inequality allows us to use the argument in [25, Section 9, Proof of Theorem 9.6, Phase III] to prove (18), (20), (19), (21), for all $t \in [t_1, \hat{t}]$. We omit the proof details. In particular, from (19), (14), and the range of $\gamma$, we have $\|X_{t,\perp}\|_* \leq \|U_\natural\|^2$.

---

[7]Note our hypotheses are the same as those in [25, (62)-(66)] with the *critical* exception (22). Here, we use the spectral norm rather than the Frobenius norm as in [25, (67)]. If we use Frobenius norm, then according to [25, Proof of Theorem 9.6, p. 27 of the supplement] and [25, Lemma 9.5], we need $\|\left(\mathcal{I} - \mathcal{A}^*\mathcal{A}\right)(\Delta_t)\|_F \leq c\kappa_f^{-2}\|X_\natural - U_t U_t^\top\|_F$. It appears very difficult (if not impossible) to justify this condition with only $2r_\natural$-RIP even if the rank of $U_t$ is no more than $r_\natural$. In the later version [52], the authors of [25] use a different method in proving [25, (66)].

Next, the inequality (17) in Lemma C.3 is satisfied due to (11). Thus we have that

$$\|V_{X_\natural}^T \Delta_{t+1}\| \leq \xi_3 \|V_{X_\natural}^T \Delta_t\| + \eta \frac{\sigma_{\min}(U_\natural)^2}{100} \|X_{t,\perp}\|_* + 18\eta \|U_\natural\|^2 \|E\|$$

$$\overset{(a)}{\leq} 10\|U_\natural\|^2 \xi_3 \xi_1^{t-t_1} + \eta \frac{\sigma_{\min}(U_\natural)^2}{100} \|X_{t,\perp}\|_* + 18\eta \|U_\natural\|^2 \|E\| \sum_{\tau=t_1+1}^{t+1} \xi_3^{\tau-t_1-1},$$

where in step (a), we use the induction hypothesis (22). Now (22) holds for $t + 1$ if the following holds.

$$\|X_{t,\perp}\|_* \leq \frac{1}{4} \xi_1^{t-t_1} \|U_\natural\|^2. \tag{23}$$

Using the relationship between the operator norm and the nuclear norm, we have

$$\|X_{t,\perp}\|_* \leq (\min\{r; n\} - r_\natural) \|U_t W_{t,\perp}\|^2$$

$$\overset{(a)}{\leq} 4(\min\{r; n\} - r_\natural)(1 + 80\eta c_2 \sigma_{\min}(U_\natural)^2)^{2(t-t_1)} \sigma_{\min}(U_\natural)^{1/4} \gamma^{7/4},$$

where in step $(a)$, we use (19) and the bound on $\|U_{t_1} W_{t_1,\perp}\|$ from (14). Hence, the inequality (23) holds if $c_2 > 0$ is small enough and $16(\min\{r; n\} - r_\natural) \sigma_{\min}(U_\natural)^{1/4} \gamma^{7/4} \leq \left(1 - \frac{\eta}{350} \sigma_{\min}(U_\natural)^2\right)^{t-t_1} \|U_\natural\|^2$. This inequality is indeed true so long as $t \leq \hat{t} = t_1 + t_{\tilde{\Delta}}$. The induction step for the case $r > r_\natural$ is finished.

Let us now verify the inequality for $\|\Delta_t\|_F$ for $r > r_\natural$:

$$\|\Delta_{\hat{t}}\|_F \overset{(a)}{\leq} 4\|V_{X_\natural}^T \Delta_{\hat{t}}\|_F + \|X_{\hat{t},\perp}\|_* \overset{(b)}{\lesssim} r_\natural^{1/2} \xi_1^{\hat{t}-t_1} \|U_\natural\|^2 + r_\natural^{1/2} \eta \|U_\natural\|^2 \|E\| \sum_{\tau=t_1+1}^{\hat{t}} \xi_3^{\tau-t_1-1}$$

$$\overset{(c)}{\lesssim} r_\natural^{1/2} \left(\frac{\kappa_f^{1/4}}{\min\{r; n\} - r_\natural} \frac{\|U_\natural\|^{7/4}}{\gamma^{7/4}}\right)^{-3/4} \|U_\natural\|^2 + r_\natural^{1/2} \|E\| \kappa_f^2 \tag{24}$$

where inequality $(a)$ follows from Lemma C.5. Inequality $(b)$ follows from (22) and (23). The step $(c)$ is due to the definition of $\hat{t}$.

**Case $r = r_\natural$:** We note that for $t = t_1$, we have $U_t = U_t W_t W_t^\top$ and $W_{t,\perp} = 0$ because $\sigma_{r_\natural}(U_t W_t) > 0$ and $W_t$ is of size $r_\natural \times r_\natural$. Following almost the same procedure as before, we can prove the induction hypothesis (18) to (22) for any $t \geq t_1$ again with (19) replaced by $\|U_t W_{t,\perp}\| = 0$. Since we can ignore the term $U_t W_{t,\perp}$ in (23), we have (22) for all $t \geq t_1$. Finally, to bound $\|U_t U_t^T - X_\natural\|_F$, we can replace $\hat{t}$ by $t$ in (24), stop at step $(b)$, and bound $r_\natural^{1/2} \eta \|U_\natural\|^2 \|E\| \sum_{\tau=t_1+1}^{\hat{t}} \left(1 - \frac{\eta}{200} \sigma_{\min}(U_\natural)^2\right)^{\tau-t_1-1}$ by $r_\natural^{1/2} \|E\| \kappa_f^2$. $\qquad \square$

## C.4. Proof of Lemma C.3

We start with the following technical lemma.

**Lemma C.5.** *Under the assumptions of Lemma C.3, the following inequalities hold:*

$$\|V_{X_\natural^\perp}^T X_t\| \leq 3\|V_{X_\natural}^T \Delta_t\| + \|X_{t,\perp}\|, \tag{25}$$

$$\|\Delta_t\| \leq 4\|V_{X_\natural}^T \Delta_t\| + \|X_{t,\perp}\|, \tag{26}$$

$$\|\tilde{\Delta}_t\| \leq 4\|V_{X_\natural}^T \Delta_t\|. \tag{27}$$

*Proof.* The first two inequalities are proved in [25, Lemma B.4]. To prove the last inequality, we first decompose $\tilde{\Delta}_t$ as

$$\tilde{\Delta}_t = V_{X_\natural} V_{X_\natural}^\top \tilde{\Delta}_t + V_{X_\natural^\perp} V_{X_\natural^\perp}^\top \tilde{\Delta}_t. \tag{28}$$

For the first term, we have

$$V_{X_\natural}V_{X_\natural}^\top\tilde{\Delta}_t \overset{(a)}{=} V_{X_\natural}V_{X_\natural}^\top X_\natural - V_{X_\natural}V_{X_\natural}^\top U_t W_t W_t^\top U_t^T$$

$$\overset{(b)}{=} V_{X_\natural}V_{X_\natural}^\top X_\natural - V_{X_\natural}V_{X_\natural}^\top U_t U_t^T = V_{X_\natural}V_{X_\natural}^\top \Delta_t.$$

Here the step $(a)$ is the definition of $\tilde{\Delta}_t$, and the step $(b)$ is due to (10). Thus, we have $\|V_{X_\natural}V_{X_\natural}^\top\tilde{\Delta}_t\| \le \|V_{X_\natural}^\top\Delta_t\|$. Our task is then to bound the second term of (28):

$$\|V_{X_\natural^\perp}V_{X_\natural^\perp}^\top\tilde{\Delta}_t\| \le \|V_{X_\natural^\perp}V_{X_\natural^\perp}^\top\tilde{\Delta}_t V_{X_\natural}\| + \|V_{X_\natural^\perp}V_{X_\natural^\perp}^\top\tilde{\Delta}_t V_{X_\natural^\perp}V_{X_\natural^\perp}^\top\|$$

$$\overset{(a)}{\le} \|\Delta_t V_{X_\natural}\| + \|V_{X_\natural^\perp}\tilde{X}_t V_{X_\natural^\perp}\|.$$

The step $(a)$ is due to (10). For the second term $\|V_{X_\natural^\perp}\tilde{X}_t V_{X_\natural^\perp}\|$, we have the following estimate:

$$\|V_{X_\natural^\perp}^T\tilde{X}_t V_{X_\natural^\perp}\| = \|V_{X_\natural^\perp}^T V_{\tilde{U}_t}V_{\tilde{U}_t}^T\tilde{X}_t V_{X_\natural^\perp}\| = \|V_{X_\natural^\perp}^T V_{\tilde{U}_t}\left(V_{X_\natural}^T V_{\tilde{U}_t}\right)^{-1}V_{X_\natural}^T V_{\tilde{U}_t}V_{\tilde{U}_t}^T\tilde{X}_t V_{X_\natural^\perp}\|$$

$$\le \|V_{X_\natural^\perp}^T V_{\tilde{U}_t}\|\|\left(V_{X_\natural}^T V_{\tilde{U}_t}\right)^{-1}\|\|V_{X_\natural}^T V_{\tilde{U}_t}V_{\tilde{U}_t}^T\tilde{X}_t V_{X_\natural^\perp}\|. \tag{29}$$

Using (10) again, we know

$$V_{X_\natural}^T V_{\tilde{U}_t}V_{\tilde{U}_t}^T\tilde{X}_t V_{X_\natural^\perp} = V_{X_\natural}^T U_t U_t^T V_{X_\natural^\perp} = V_{X_\natural}^T\Delta_t V_{X_\natural^\perp}.$$

We also have $\dfrac{\|V_{X_\natural^\perp}^T V_{\tilde{U}_t}\|}{\sigma_{\min}\left(V_{X_\natural}^T V_{\tilde{U}_t}\right)} \le 2$ due to $\|V_{X_\natural^\perp}^\top V_{\tilde{U}_t}\| \le c\kappa_f^{-2}$. This is true by noting that $V_{\tilde{U}_t} = $

$V_{X_\natural}V_{X_\natural}^\top V_{\tilde{U}_t} + V_{X_\natural^\perp}V_{X_\natural^\perp}^\top V_{\tilde{U}_t}$ and $\sigma_{r^\star}(V_{X_\natural}^\top V_{\tilde{U}_t}) = \sigma_{r^\star}(V_{X_\natural}V_{X_\natural}^\top V_{\tilde{U}_t}) \overset{(a)}{\ge} \sigma_{r^\star}(V_{\tilde{U}_t}) - \|V_{X_\natural^\perp}V_{X_\natural^\perp}^\top V_{\tilde{U}_t}\| = 1 - \|V_{X_\natural^\perp}^\top V_{\tilde{U}_t}\|$, where the step $(a)$ is due to Weyl's inequality. Thus, the proof is completed by continuing the chain of inequality of (29): $\|V_{X_\natural^\perp}^T\tilde{X}_t V_{X_\natural^\perp}\| \le \dfrac{\|V_{X_\natural^\perp}^T V_{\tilde{U}_t}\|}{\sigma_{\min}\left(V_{X_\natural}^T V_{\tilde{U}_t}\right)}\|V_{X_\natural}^T\Delta_t V_{X_\natural^\perp}\| \le 2\|V_{X_\natural}^T\Delta_t\|$.  $\square$

Let us now prove Lemma C.3.

*Proof of Lemma C.3.* As in [25, Proof of Lemma 9.5], we can decompose $\Delta_{t+1} = X_\natural - X_{t+1}$ into five terms by using the formula $U_{t+1} = U_t + \eta\left[(\mathcal{A}^*\mathcal{A})(\Delta_t) + E\right]U_t$ and $X_{t+1} = U_{t+1}U_{t+1}^\top$:

$$\Delta_{t+1} = \underbrace{(I - \eta X_t)(\Delta_t)(I - \eta X_t)}_{=Q_1} + \eta\underbrace{\left[(\mathcal{I} - \mathcal{A}^*\mathcal{A})(\Delta_t) + E\right]X_t}_{=Q_2}$$

$$+ \eta\underbrace{X_t\left[(\mathcal{I} - \mathcal{A}^*\mathcal{A})(\Delta_t) + E\right]}_{=Q_3} - \eta^2\underbrace{X_t\Delta_t X_t}_{=Q_4}$$

$$- \eta^2\underbrace{\left[(\mathcal{A}^*\mathcal{A})(\Delta_t) + E\right]X_t\left[(\mathcal{A}^*\mathcal{A})(\Delta_t) + E\right]}_{=Q_5}.$$

Here, $I$ is the identity matrix. We shall bound $V_{X_\natural}^T Q_i$, $i = 1, \ldots, 5$. Their bounds below and that $\eta \le c\kappa_f^{-2}\|U_\natural\|^{-2}$ give the result.

**Bounding $V_{X_\natural}^T Q_1$ and $V_{X_\natural}^T Q_4$:** Because these two terms do not involve the noise matrix $E$, we may recycle the proof of [25, Lemma 9.5, Estimation of (I) and (IV)] and conclude that

$$\|V_{X_\natural}^T Q_1\| \le (1 - \frac{\eta}{40}\sigma_{\min}^2(U_\natural))\|V_{X_\natural}^\top\Delta_t\| + \eta\frac{\sigma_{\min}^2(U_\natural)}{400}\|X_{t,\perp}\|$$

and

$$\eta^2\|V_{X_\natural}^\top X_t(\Delta_t)X_t^\top\| \le \frac{\eta}{1000}\sigma_{\min}^2(U_\natural)\left(5\|V_{X_\natural}^T\Delta_t\| + \|X_{t,\perp}\|\right).$$

**Bounding $V_{X_\natural}^\top Q_2$:** From triangle inequality and submultiplicity of $\|\cdot\|$, we have

$$\|V_{X_\natural}^T \left[(\mathcal{I} - \mathcal{A}^*\mathcal{A})(\Delta_t) + E\right] U_t U_t^T\|$$

$$\leq \left(\|(\mathcal{I} - \mathcal{A}^*\mathcal{A})(\Delta_t)\| + \|V_X^\top E\|\right)\|U_t\|^2$$

$$\overset{(a)}{\leq} 3^2 \left(\|(\mathcal{I} - \mathcal{A}^*\mathcal{A})(\Delta_t)\| + \|V_X^\top E\|\right)\|U_\natural\|^2$$

$$\overset{(b)}{\leq} 3^3 c\sigma_{\min}^2(U_\natural)\left(\|X_\natural - U_t W_t W_t^\top U_t^T\| + \|U_t W_{t,\perp} W_{t,\perp}^\top U_t\|_*\right)$$
$$\qquad + 9\|V_{X_\natural}^\top E\|\|U_\natural\|^2$$

$$\overset{(c)}{\leq} 4 \cdot 3^3 c\sigma_{\min}^2(U_\natural)\left(\|V_{X_\natural}^T(\Delta_t)\| + \|U_t W_{t,\perp} W_{t,\perp}^T U_t\|_*\right)$$
$$\qquad + 3^2\|V_{X_\natural}^\top E\|\|U_\natural\|^2.$$

In the step $(a)$, we use the assumption $\|U\| \leq 3\|U_\natural\|$, and in the step $(b)$, we use assumption (17). In the step $(c)$, we use Lemma C.5. By taking a small constant $c > 0$, we have

$$\|V_{X_\natural}^T\left[(\mathcal{I} - \mathcal{A}^*\mathcal{A})(X_\natural - UU_t^T)\right]U_t U_t^T\|$$

$$\leq \frac{1}{1000}\sigma_{\min}^2(U_\natural)\left(\|V_{X_\natural}^T\Delta_t\| + \|X_{t,\perp}\|_*\right) + 3^2\|V_{X_\natural}^\top E\|\|U_\natural\|^2.$$

**Bounding $V_{X_\natural} Q_3$:** We derive the following inequality similarly as bounding $V_{X_\natural} Q_3$:

$$\|V_{X_\natural}^T U_t U_t^T\left[(\mathcal{I} - \mathcal{A}^*\mathcal{A})(\Delta_t) + E\right]\|$$

$$\leq \frac{1}{1000}\sigma_{\min}^2(U_\natural)\left(\|V_{X_\natural}^T\Delta_t\| + \|X_{t,\perp}\|_*\right) + 3^2\|E\|\|U_\natural\|^2.$$

**Bounding $V_{X_\natural} Q_5$:** First, we have the following bound:

$$\|(\mathcal{A}^*\mathcal{A})(\Delta_t)\| \leq \|\Delta_t\| + \|\left[(\mathcal{I} - \mathcal{A}^*\mathcal{A})(\Delta_t)\right]\|$$

$$\leq 2\left(\|\tilde{\Delta}_t\| + \|X_{t,\perp}\|_*\right), \tag{30}$$

$$\leq 2\left(2\|U_\natural\|^2 + \|U_t\|^2\right). \tag{31}$$

In the step (30), we use the assumption (17). In the step (31), we use the assumptions (17) and $\|X_{t,\perp}\|_* \leq \|U_\natural\|^2$. We can bound the term $V_{X_\natural} Q_5$ as follows:

$$\|V_{X_\natural}^T\left[(\mathcal{A}^*\mathcal{A})(\Delta_t) + E\right]U_t U_t^T\left[(\mathcal{A}^*\mathcal{A})(\Delta_t) + E\right]\|$$

$$\leq \left(\|\left[(\mathcal{A}^*\mathcal{A})(\Delta_t)\right]\| + \|E\|\right)\|U_t\|^2\left(\|(\mathcal{A}^*\mathcal{A})(\Delta_t)\| + \|E\|\right)$$

$$\overset{(a)}{\leq} 4\left(\|\tilde{\Delta}_t\| + \|X_{t,\perp}\|_* + \|E\|\right)\|U_t\|^2\left(2\|U_\natural\|^2 + \|U_t\|^2 + \|E\|\right)$$

$$\overset{(b)}{\leq} 6^4\left(\|\tilde{\Delta}_t\| + \|\tilde{X}_t\|_* + \|E\|\right)\|U_\natural\|^4$$

$$\overset{(c)}{\leq} 4 \cdot 6^4\left(\|V_{X_\natural}^T\Delta_t\| + \|X_{t,\perp}\|_* + \|E\|\right)\|U_\natural\|^4.$$

Here, in the step $(a)$, we used the (30) and (31). The step $(b)$ is due to the assumption $\|U_t\| \leq 3\|U_\natural\|$ and $\|E\| \leq \|U_\natural\|^2$. In the step $(c)$, we used Lemma C.5. Thus, we have

$$\eta^2\|V_{X_\natural}^T Q_5\| \overset{(a)}{\leq} \frac{\eta\sigma_{\min}^2(X)}{1000}\left(\|V_{X_\natural}^T\Delta_t\| + 2.5\|X_{t,\perp}\|_* + 2.5\|E\|\right),$$

where the step $(a)$ is due to the assumption on the stepsize $\eta \leq c\kappa_f^{-2}\|U_\natural\|^{-2}$.

$\square$

# D. Proof in Section 3

We start with the proof of Lemma D.1.

To get a guarantee of the validation approach, we start with a lemma showing that the validation error of an arbitrary sequence $D_1, \ldots, D_T$ is close to its expectation if the entries of $A_i$ are iid drawn from a subgaussian distribution.

**Lemma D.1.** *Suppose each entry in $A_i$, $i \in \mathcal{I}_{\mathrm{val}}$ is i.i.d.* $\mathrm{subG}(c_1)$ *with mean zero and variance 1 and each $e_i$ is also a zero-mean sub-Gaussian distribution* $\mathrm{subG}(c_2 \sigma^2)$ *with variance $\sigma^2$, where $c_1, c_2 \geq 1$ are absolute constants. Let $T > 0$. Assume matrices $D_1, \ldots, D_T \in \mathbb{R}^{d \times d}$ are independent of $\mathcal{A}_{\mathrm{val}}$ and $e_{\mathrm{val}}$. For any $\delta_{\mathrm{val}} > 0$, if $m_{\mathrm{val}} \geq C_1 \frac{\log T}{\delta_{\mathrm{val}}^2}$, then with probability at least $1 - 2T \exp\left(-C_2 m_{\mathrm{val}} \delta_{\mathrm{val}}^2\right)$,*

$$\left| \|\mathcal{A}_{\mathrm{val}}(D_t) + e\|_2^2 - m_{\mathrm{val}}(\|D_t\|_F^2 + \sigma^2) \right| \geq \delta_{\mathrm{val}} m_{\mathrm{val}}(\|D_t\|_F^2 + \sigma^2), \forall\, t = 1, \ldots, T. \tag{32}$$

*Here $C_1, C_2 \geq 0$ are constants that only depend on $c_1$ and $c_2$.*

*Proof.* For any $D \in \mathbb{R}^{n \times n}$, $\langle A_i, D \rangle + e_i$ is a zero-mean sub-Gaussian random variable with variance $\|D\|_F^2 + \sigma^2$ and sub-Gaussian norm

$$\|\langle A_i, D \rangle + e_i\|_{\psi_2}^2 \leq C \left(c_1 \|D\|_F^2 + c_2 \sigma^2\right) \leq C_1 \left(\|D\|_F^2 + \sigma^2\right),$$

where $C$ is an absolute constant and $C_1 \geq 0$ is a constant depending on $c_1$ and $c_2$. The above inequality implies that $(\langle A_i, D \rangle + e_i)^2$ is sub-exponential with a sub-exponential norm the same as $\|\langle A_i, D \rangle + e_i\|_{\psi_2}^2$ [53, Theorem 2.7.6]. For each $s \geq 0$ and $t = 1, \ldots, T$, define the event $S_t(s)$ to be

$$\left\{ \left| \frac{1}{m_{\mathrm{val}}} \|\mathcal{A}_{\mathrm{val}}(D_t) + e\|_F^2 - (\|D_t\|_F^2 + \sigma^2) \right| \geq s C_1 (\|D\|_F^2 + \sigma^2) \right\}$$

Using [53, Theorem 2.8.1], we have $\mathbb{P}\left(S_t(s)\right) \leq 2 \exp\left(-C_2 \min\left(m_{\mathrm{val}} s^2, m_{\mathrm{val}} s\right)\right)$, where $C_2 > 0$ is an absolute constant. Taking $s = \delta_{\mathrm{val}}/C_1$ and $m_{\mathrm{val}} = O(\frac{\log^2 T}{\delta_{\mathrm{val}}^2})$, the union bound for $S_1(s), \ldots, S_T(s)$ implies that

$$\mathbb{P}(\cup_{t=1}^T S_t(s)) \leq 2T \exp\left(-C_2 m_{\mathrm{val}} \delta_{\mathrm{val}}^2\right),$$

where we absorb $C_1$ into $C_2$ in the last equation. $\square$

Next, we show the error identified by the validation is close to the optimal if each validation error is close to its expectation.

**Lemma D.2.** *Given any $T > 0$ and matrices $D_1, \ldots, D_T \in \mathbb{R}^{d \times d}$, define $\hat{t} = \arg\min_{1 \leq t \leq T} \|\mathcal{A}_{\mathrm{val}}(D_t) + e\|_2$ and $\tilde{t} = \arg\min_{1 \leq t \leq T} \|D_t\|_F$. If*

$$(1 - \delta_{\mathrm{val}})(\|D_t\|_F^2 + \sigma^2) \leq \frac{1}{m_{\mathrm{val}}} \|\mathcal{A}_{\mathrm{val}}(D_t) + e\|_2^2 (1 + \delta_{\mathrm{val}})(\|D_t\|_F^2 + \sigma^2), \forall t = 1, \ldots, T, \tag{33}$$

*then,*

$$\|D_{\hat{t}}\|_F^2 \leq \frac{1 + \delta_{\mathrm{val}}}{1 - \delta_{\mathrm{val}}} \|D_{\tilde{t}}\|_F^2 + \frac{2\delta_{\mathrm{val}}}{1 - \delta_{\mathrm{val}}} \sigma^2.$$

*Proof of Lemma D.2.* Since $\hat{t} = \arg\min_{1 \leq t \leq T} \|\mathcal{A}(D_t) + e\|_2$, we have

$$\|D_{\hat{t}}\|_F^2 + \sigma^2 \leq \frac{1}{1 - \delta_{\mathrm{val}}} \|\mathcal{A}_{\mathrm{val}}(D_{\hat{t}}) + e\|_F^2$$

$$\leq \frac{1}{1 - \delta_{\mathrm{val}}} \|\mathcal{A}_{\mathrm{val}}(D_{\tilde{t}}) + e\|_F^2 \leq \frac{1 + \delta_{\mathrm{val}}}{1 - \delta_{\mathrm{val}}}(\|D_{\tilde{t}}\|_F^2 + \sigma^2),$$

which further implies that

$$\|D_{\hat{t}}\|_F^2 \leq \frac{1 + \delta_{\mathrm{val}}}{1 - \delta_{\mathrm{val}}} \|D_{\tilde{t}}\|_F^2 + \frac{2\delta_{\mathrm{val}}}{1 - \delta_{\mathrm{val}}} \sigma^2.$$

$\square$

Finally, let us prove Theorem 3.1.

*Proof of Theorem 3.1.* Let $\delta_{\mathrm{val}} = \frac{\kappa^2 n r_\natural}{m_{\mathrm{train}}}$ and $D_t = U_t U_t^\top - X_\natural$ where $U_t, t = 1, \ldots, T$ are iterates from (3) with $(y, \mathcal{A})$ replaced by $(y_{\mathrm{train}}, \mathcal{A}_{\mathrm{train}})$. With this choice of $\delta_{\mathrm{val}}$ and $D_t$ and the above lemmas, the theorem is immediate. $\square$

# E. Additional Experiments on DIP

In Section 4, we present the validity of our hold-out method for determining the denoised image through the training progress across different types of noise. In this section, we further conduct extensive experiments to investigate and verify the universal effectiveness. In Section E.1, we demonstrate that validity of our method is not limited on Adam by the experiments on SGD. In Section E.2, we demonstrate that validity of our method also holds for L1 loss function. In Section E.3, we demonstrate that validity of our method also exists across a wide range of noise degree for both gaussian noise and salt and pepper noise.

## E.1. Validty across different optimizers

In this part, we verify the success of our method exists across different types of optimizers for dfferent noise. We use Adam with learning rate $0.05$ (at the top row) and SGD with learning rate $5$ (at the bottom row) to train the network for recovering the noise images under L2 loss for 20000 iterations, where we evaluate the validation loss between generated images and corrupted images , and the PSNR between generated image and clean images every 200 iterations. In Figure 4, we show that both Adam and SGD work for Gaussian noise, where we use the Gaussian noise with mean 0 and variance 0.2. At the top row, we plot the results of recovering images optimized by Adam. The PSNR of chosen recovered image according to the validation loss is $20.7518$ at the second column, which is comparable to the Best PSNR $20.8550$ through the training progress at the third column. The best PSNR of full noisy image is $21.0714$ through the training progress at the last column. At the bottom row, we show the results of SGD, The PSNR of decided image according to the validation loss is $20.6964$ at the second column, which is comparable to the Best PSNR $20.7551$ through the training progress at the third column. The best PSNR of full noisy image is $20.9449$ through the training progress at the last column. In Figure 5, we show that both Adam and SGD works for salt and pepper noise, where $10\%$ percent of pixels are randomly corrupted. At the top row, we plot the results of recovering images optimized by Adam. The PSNR of chosen recovered image according to the validation loss is $20.8008$ at the second column, which is comparable to the Best PSNR $20.9488$ through the training progress at the third column. The best PSNR of full noisy image is $21.0576$ through the training progress at the last column. At the bottom row, we show the results of SGD, The PSNR of decided image according to the validation loss is $20.7496$ at the second column, which is comparable to the Best PSNR $20.8691$ through the training progress at the third column. The best PSNR of full noisy image is $21.0461$ through the training progress at the last column. Comparing these results, the SGD optimization algorithm usually takes more iterations to recovery the noisy images corrupted by either Gaussian noise and salt and pepper noise, therefore we will use Adam in the following parts.

## E.2. Validty across loss functions

In this part, we verify the success of our method exists across different types of loss functions for different noise. We use Adam with learning rate $0.05$ to train the network for recovering the noise images under either L1 loss (at the top row) or L2 loss (at the bottom row) for 50000 iterations, where we evaluate the validation loss between generated images and corrupted images, and the PSNR between generated image and clean images every 500 iterations. In Figure 6, we show that both L1 loss and L2 loss works for Gaussian noise, where we use the Gaussian noise with mean 0 and variance 0.2. At the top row, we plot the results of recovering images under L1 loss. The PSNR of chosen recovered image according to the validation loss is $27.5232$ at the second column, which is comparable to the Best PSNR $27.5233$ through the training progress at the third column. The best PSNR of full noisy image is $28.1647$ through the training progress at the last column. At the bottom row, we show the results under L2 loss, The PSNR of decided image according to the validation loss is $27.7941$ at the second column, which is comparable to the Best PSNR $27.9268$ through the training progress at the third column. The best PSNR of full noisy image is $27.9370$ through the training progress at the last column. In Figure 7, we show that both L1 loss and L2 loss works for salt and pepper noise, where $30\%$ percent of pixels are randomly corrupted. At the top row, we plot

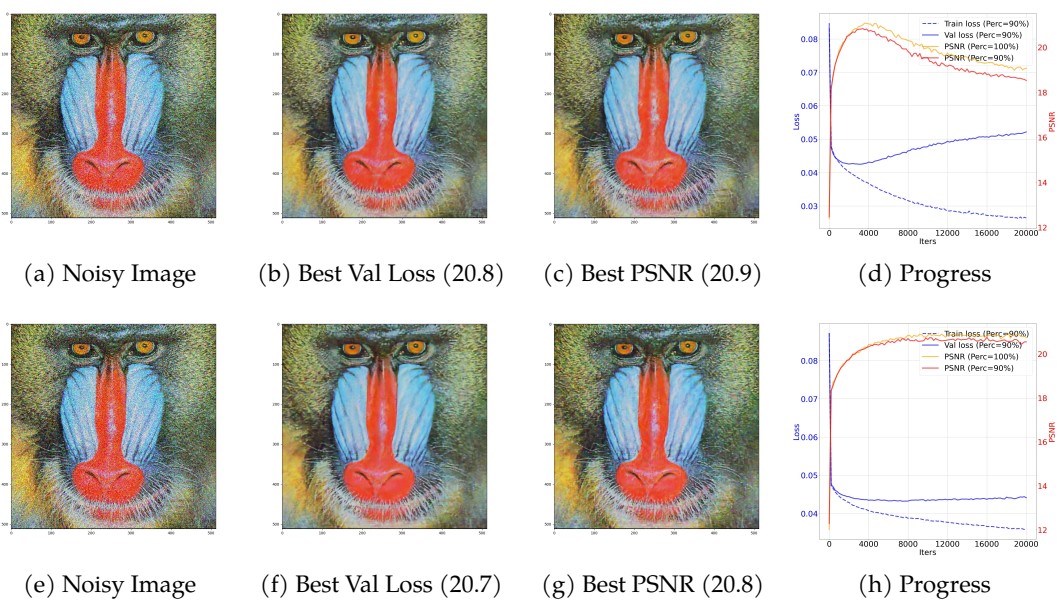

(a) Noisy Image     (b) Best Val Loss (20.8)     (c) Best PSNR (20.9)     (d) Progress

(e) Noisy Image     (f) Best Val Loss (20.7)     (g) Best PSNR (20.8)     (h) Progress

Figure 4: **Results across different optimizer for guassian noise**. The top row plots the results for Adam, and the bottom row plots the results for SGD.

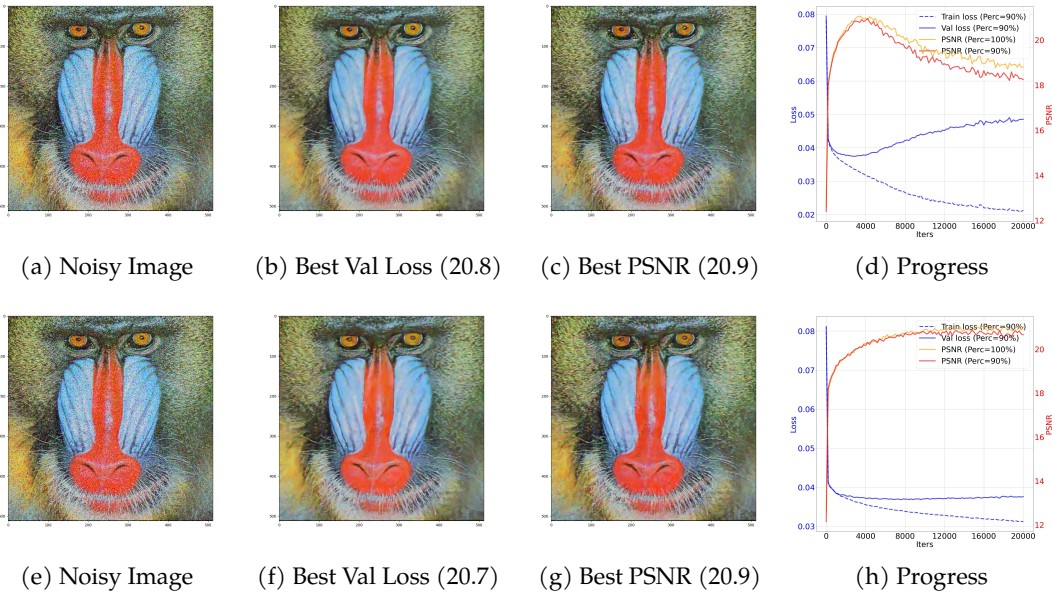

(a) Noisy Image     (b) Best Val Loss (20.8)     (c) Best PSNR (20.9)     (d) Progress

(e) Noisy Image     (f) Best Val Loss (20.7)     (g) Best PSNR (20.9)     (h) Progress

Figure 5: **Results across different optimizer for salt and pepper noise**. The top row plots the results for Adam, and the bottom row plots the results for SGD.

the results of recovering images under L1 loss. The PSNR of chosen recovered image according to the validation loss is 35.2155 at the second column, which is comparable to the Best PSNR 35.2266 through the training progress at the third column. The best PSNR of full noisy image is 35.9319 through the training progress at the last column. At the bottom row, we show the results under L2 loss, The PSNR of decided image according to the validation loss is 21.8679 at the second column, which is comparable to the Best PSNR 21.8679 through the training progress at the third column. The best PSNR of full noisy image is 22.1088 through the training progress at the last column. Comparing these results, the L1 loss usually performs comparably as L2 loss to recovery the noisy images

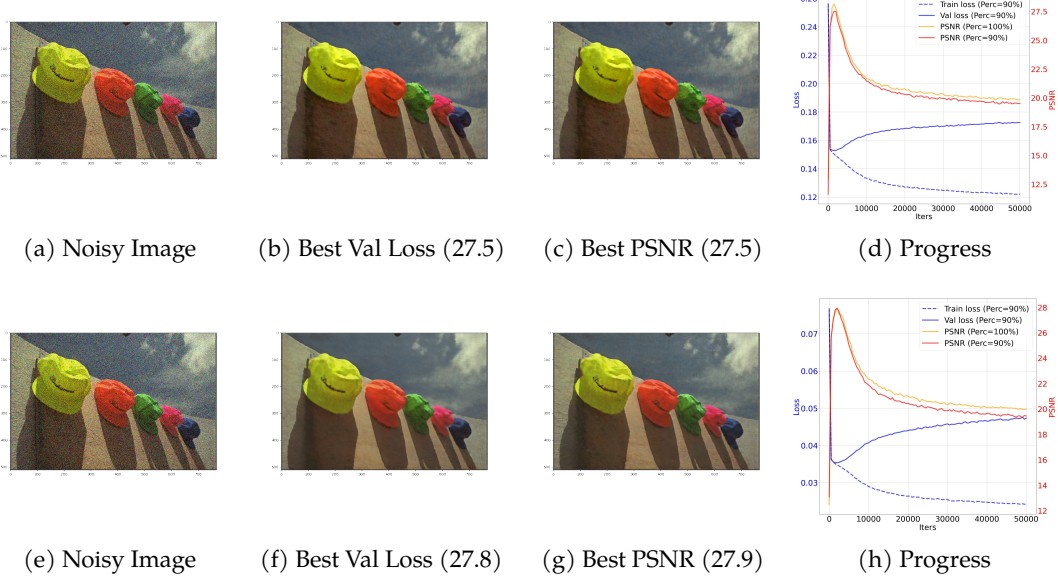

(a) Noisy Image     (b) Best Val Loss (27.5)     (c) Best PSNR (27.5)     (d) Progress

(e) Noisy Image     (f) Best Val Loss (27.8)     (g) Best PSNR (27.9)     (h) Progress

Figure 6: **Results across different losses for guassian noise**. The top row plots the results for L1 loss, and the bottom row plots the results for L2 loss.

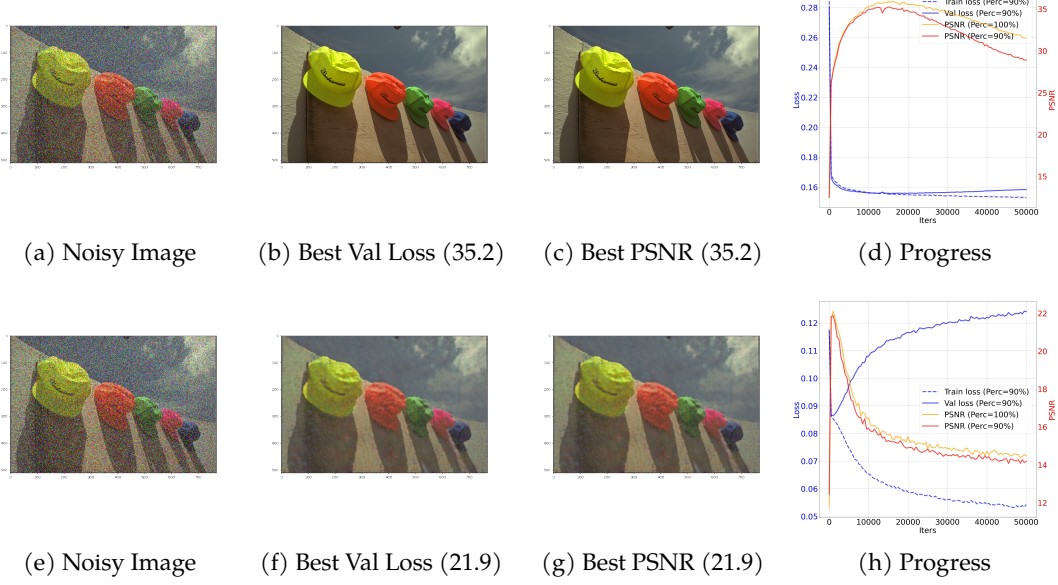

(a) Noisy Image     (b) Best Val Loss (35.2)     (c) Best PSNR (35.2)     (d) Progress

(e) Noisy Image     (f) Best Val Loss (21.9)     (g) Best PSNR (21.9)     (h) Progress

Figure 7: **Results across different losses for salt and pepper noise**. The top row plots the results for L1 loss, and the bottom row plots the results for L2 loss.

corrupted by either Gaussian noise, and L1 loss produces cleaner recovered image than L2 loss for salt and pepper noise, therefore, we will use L1 loss in the following parts.

### E.3. Validty across noise degree

In this part, we verify the success of our method exists across different noise degrees for different noise. We use Adam with learning rate $0.05$ to train the network for recovering the noise images under L1 loss for 50000 iterations, where we evaluate the validation loss between generated images and

corrupted images, and the PSNR between generated image and clean images every 500 iterations. In Figure 8, we show that our methods works for different noise degree of Gaussian noise, where we use the Gaussian noise with mean 0. At the top row, we plot the results of recovering images under 0.1 variance of gaussian noise. The PSNR of chosen recovered image according to the validation loss is 28.6694 at the second column, which is comparable to the Best PSNR 28.6694 through the training progress at the third column. The best PSNR of full noisy image is 28.9929 through the training progress at the last column. At the middel row, we plot the results of recovering images under 0.2 variance of gaussian noise. The PSNR of chosen recovered image according to the validation loss is 25.6036 at the second column, which is comparable to the Best PSNR 25.6839 through the training progress at the third column. The best PSNR of full noisy image is 25.8879 through the training progress at the last column. At the bottom row, we show the results under under 0.3 variance of gaussian noise, The PSNR of decided image according to the validation loss is 23.7784 at the second column, which is identical to the Best PSNR 23.7784 through the training progress at the third column. The best PSNR of full noisy image is 23.9001 through the training progress at the last column. In Figure 9, we show that our methods works for different noise degree of salt and pepper noise. At the top row, we plot the results of recovering images with 10 percent of pixels randomly corrupted by salt and pepper noise. The PSNR of chosen recovered image according to the validation loss is 35.3710 at the second column, which is comparable to the Best PSNR 35.5430 through the training progress at the third column. The best PSNR of full noisy image is 35.9778 through the training progress at the last column. At the middel row, we plot the results of recovering images under 30 percent of pixels randomly corrupted by salt and pepper noise. The PSNR of chosen recovered image according to the validation loss is 32.9904 at the second column, which is comparable to the Best PSNR 33.0188 through the training progress at the third column. The best PSNR of full noisy image is 33.4267 through the training progress at the last column. At the bottom row, we show the results under 50 percent of pixels randomly corrupted by salt and pepper noise The PSNR of decided image according to the validation loss is 29.8836 at the second column, which is identical to the Best PSNR 29.8836 through the training progress at the third column. The best PSNR of full noisy image is 30.1512 through the training progress at the last column. Comparing both the results of gaussian noise and salt and pepper noise, we can draw three conclusion. First, the PSNR of recovered image drops with the noise degree increases. Second, The peak of PSNR occurs earlier with the noise degree increases. Last, the neighbor near the peak of PSNR becomes shaper with the noise degree increases.

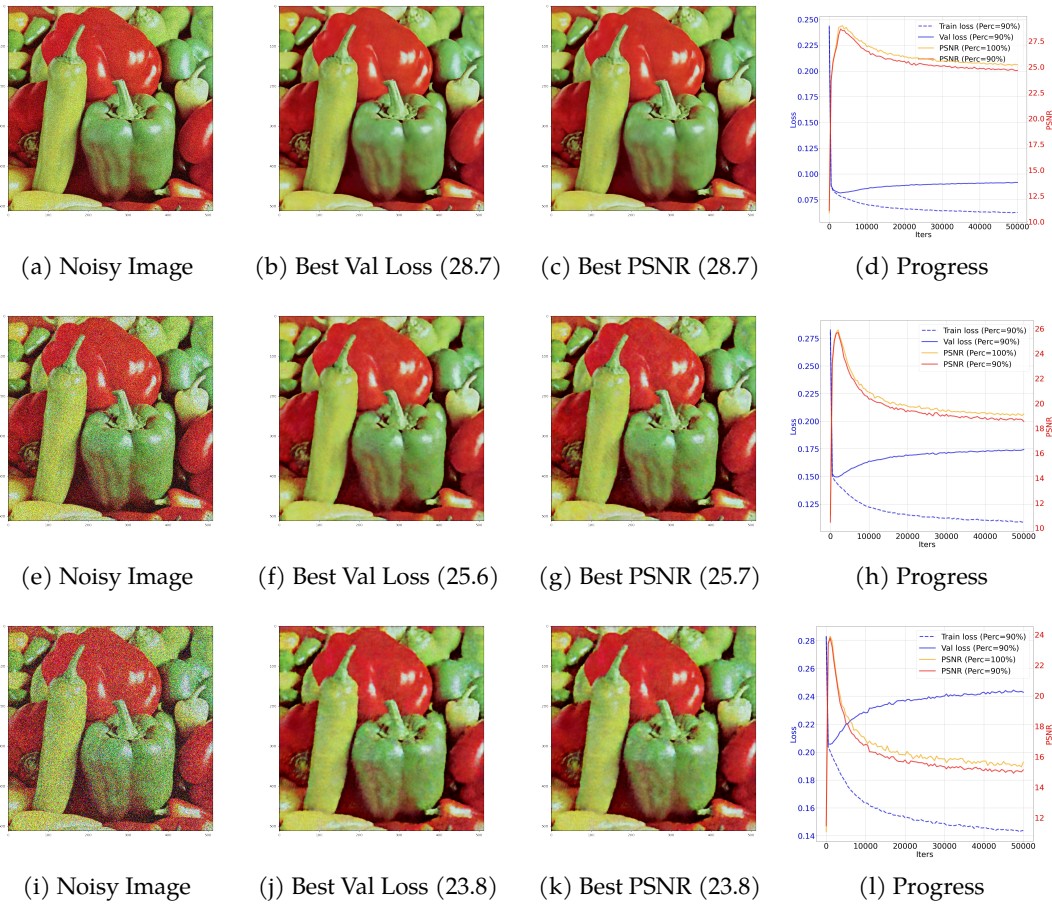

Figure 8: **Results across noise degree for gaussian noise**. The top row plots the results for gaussian noise with mean 0 and variance 0.1, the middle row plots the results for gaussian noise with mean 0 and variance 0.2, and the bottom row plots the result for gaussian noise with mean 0 and variance 0.3.

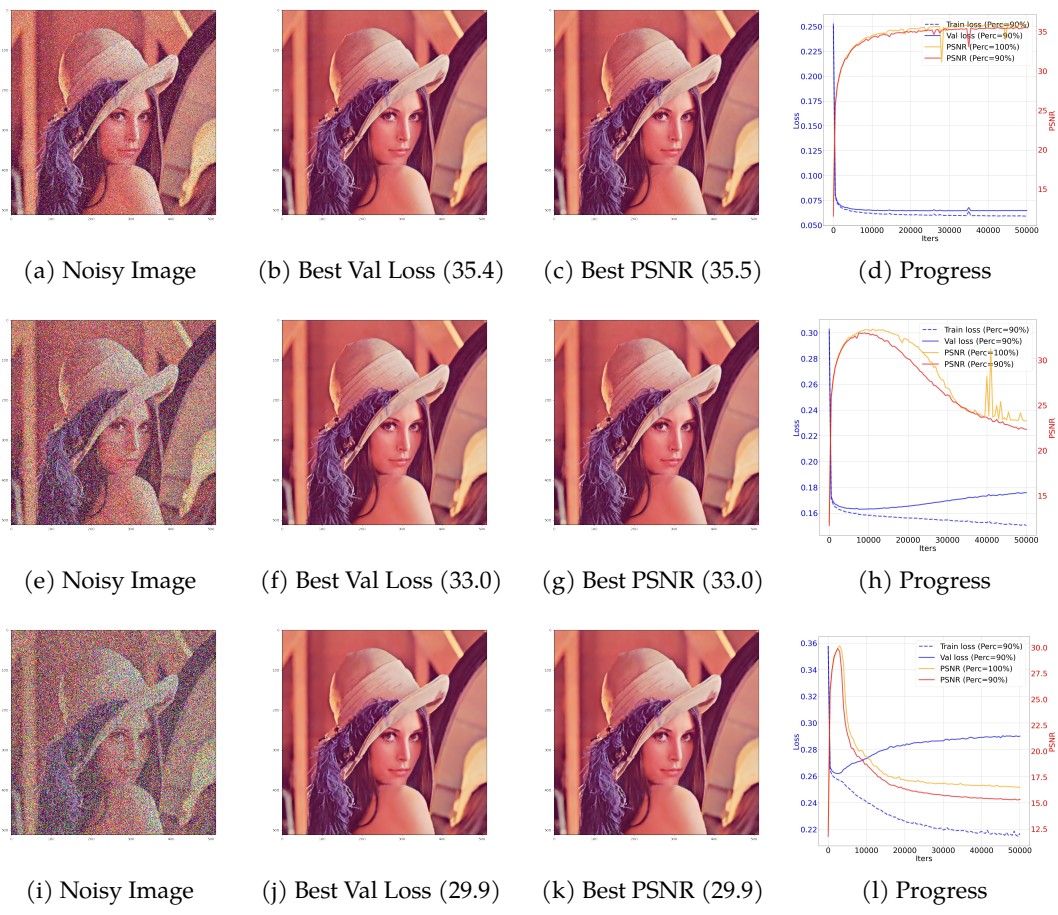

(a) Noisy Image     (b) Best Val Loss (35.4)     (c) Best PSNR (35.5)     (d) Progress

(e) Noisy Image     (f) Best Val Loss (33.0)     (g) Best PSNR (33.0)     (h) Progress

(i) Noisy Image     (j) Best Val Loss (29.9)     (k) Best PSNR (29.9)     (l) Progress

Figure 9: **Results across noise degree for salt and pepper noise**. The top row plots the results for 10 percent of corrupted pixels, the middle row plots the results for 30 percent of corrupted pixels, and the bottom row plots the result for 50 percent of corrupted pixels.

