# OpenReview forum: "A Validation Approach to Over-parameterized Matrix and Image Recovery"
_CPAL.cc/2025/Proceedings_Track — CPAL 2025 (Proceedings Track) Poster_

### Official Review · Reviewer_RLTh · 2025-01-03

**Rating:** 8
**Confidence:** 4

**Review:**

This paper introduces a validation approach to over-parameterized matrix and image recovery from noisy linear measurements. It shows that gradient descent with small random initialization, when stopped using a hold-out validation set, achieves nearly statistically optimal recovery, even when the parametrized rank significantly exceeds the true rank.  The paper elegantly addresses the crucial issue of overfitting in over-parameterized models, bridging the gap between computational efficiency and statistical optimality. The theoretical analysis is rigorous and provides valuable insights into the dynamics of gradient descent in this setting. Practically, the authors also demonstrated effective recovery in standard image denoising tasks, which supports the theoretical findings.

Overall the reviewer finds this paper very solid, hence recommending acceptance at CPAL.

---

### Official Review · Reviewer_DpC5 · 2025-01-12
**A theoretical analysis of gradient descent for noisy over-parameterized matrix recovery**

**Rating:** 6
**Confidence:** 4

**Review:**

This paper investigates gradient descent (GD) for low-rank matrix recovery from noisy random linear measurements. The study addresses the scenario where the rank of the ground-truth matrix is unknown, using an over-parameterized factored representation for the matrix. While such representations can lead to overfitting by fitting noise, the authors demonstrate that GD iterations with small random initialization (SRI) can achieve nearly information-theoretic optimal recovery when terminated appropriately. They propose a straightforward stopping criterion based on the widely-used hold-out method and show that it enables effective matrix recovery and completion. Additionally, the authors extend the proposed stopping strategy to deep image prior-based image reconstruction, achieving promising results.

The theory of the paper is convincing the proposed stopping criteria seems effective. However, this manuscript requires a substantial revision to improve the clarity and the experiments. Please address the following items:
1. The introduction is overly lengthy and difficult to follow. Furthermore, the contributions of the paper are not clearly and concretely presented. The authors are strongly encouraged to shorten the introduction and explicitly present the key contributions of the paper in a clear and concise manner.
2. The authors claim that a small random initialization can prevent overfitting when gradient descent (GD) iterations are appropriately stopped. This small random initialization can be interpreted as a strong prior, acting to regularize the learning trajectory. Could this prior be integrated with other prior models through Equation (2)? An exploration of this combination could provide deeper insights and expand the applicability of the method.
3. While the experimental results seem promising, the absence of baseline comparisons weakens the impact of the findings. Incorporating performance comparisons with established baselines is essential to support the contributions outlined in the abstract and introduction. For instance, the following reference can be considered:

[1] Heckel, Reinhard, and Paul Hand. "Deep decoder: Concise image representations from untrained non-convolutional networks." ICLR 2018.

---

### Official Review · Reviewer_Zx8z · 2025-01-16
**Validation-Driven Recovery of Low-Rank Matrices**

**Rating:** 7
**Confidence:** 3

**Review:**

The paper studies the problem of recovering low-rank matrices from noisy linear measurements when the true rank is unknown. The authors formulate the optimization problem over a factored representation $X=UU^T$, with $U\in \mathbb{R}^{n\times r}$. They consider an overparameterization ($r>r_{\natural}$) to account for the fact that the ground-truth rank is not known in advance.
Existing approaches that apply gradient descent with spectral initialization face two key issues: (1) they require strong RIP conditions on the measurement operators, and (2) their test error scales linearly with $r$, rather than with the true rank $r_\natural$.

To overcome these shortcomings, the paper proposes a new method that combines gradient descent (GD) with small random initialization (SRI) and introduces a validation-based stopping strategy to prevent overfitting. This approach circumvents the challenges posed by spectral initialization.

The authors provide a theoretical analysis showing that GD with SRI achieves a nearly minimax-optimal recovery error under weaker RIP conditions, specifically requiring only $2r_\natural$-RIP rather than $2r$-RIP. Additionally, they demonstrate that the validation scheme successfully identifies an iterate that achieves this minimax-optimal error rate.
They also empirically demonstrate the effectiveness of their validation approach by applying it to image denoising tasks using the deep image prior.

**Strengths:**
- The paper makes important contributions by highlighting, both theoretically and through numerical experiments, the advantages of using small random initialization for the matrix recovery problem.
- The proposed validation approach is supported by theoretical analysis and is empirically shown to effectively address overfitting.
- The paper is well-written and follows a clear, coherent structure.

**Weaknesses:** I do not find any major weaknesses in the submission.

---

### Meta-Review · Area_Chair_4Gf5 · 2025-02-04

**Recommendation:** Accept (Poster)
**Confidence:** 4

**Metareview:**

This paper addresses the problem of recovering low-rank matrices from noisy linear measurements when the true rank is unknown, proposing a method that combines gradient descent with small random initialization and a validation-based stopping strategy. The theoretical analysis demonstrates minimax-optimal recovery under weaker RIP conditions, and the empirical results further validate its effectiveness in both matrix recovery and image denoising tasks. While minor revisions could improve clarity and experimental comparisons, the work makes significant contributions with rigorous theoretical insights and practical relevance. I thus recommend acceptance.

---

### Decision · Program_Chairs · 2025-02-11

Accept (Poster)